# EmWorld: Emotion World Model with Latent State Evolution for Scenario-Incremental Dynamic Facial Expression Recognition

**Ke Wang** [1]   **Yuanyuan Liu** [* 1]   **Kejun Liu** [1]   **Yuyang Xia** [1]   **Chang Tang** [2]   **Yibing Zhan** [3]   **Zhe Chen** [4]

## Abstract

Dynamic Facial Expression Recognition (DFER) models the temporal evolution of facial expressions in videos. In real-world scenarios, changing scenarios distort expression trajectories, challenging existing methods. Most current approaches address this via passive feature alignment or domain-incremental learning but do not explicitly model scenario evolution, limiting their ability to capture expression dynamics under scenario-incremental changes. To address this, we propose **EmWorld**, an emotion world model for DFER that explicitly models latent emotion state evolution under scenario variations. Specifically, **EmWorld** formulates scenario-incremental DFER as a progressive Bayesian inference problem over latent world states with dual temporal scales. Slow-timescale component (**STS**) models scenario evolution using stochastic evolutionary priors, capturing long-term scenario effects and providing proactive guidance in new scenarios. Fast-timescale component (**FTS**) models frame-level expression dynamics with temporally consistent latent transitions, decoupling expression dynamics from scenario influences. By jointly inferring latent states at both timescales, EmWorld shifts DFER from a passive feature discrimination to active probabilistic state inference under evolving scenarios. Experiments on FERV39k, DFEW, and MAFW demonstrate that EmWorld consistently outperforms state-of-the-art methods, achieving up to 3.84% improvement while exhibiting strong cross-scenario stability and long-term robustness.

[1]School of Computer Science, China University of Geosciences (Wuhan), Wuhan, China. [2]School of Software, Huazhong University of Science and Technology, Wuhan, China. [3]School of Computer Science, Wuhan University, Wuhan, China. [4]School of Computing, Engineering and Mathematical Sciences, La Trobe University, Melbourne, Australia. Correspondence to: **\*Yuanyuan Liu (Corresponding author)** <liuyy@cug.edu.cn>.

*Proceedings of the $43^{rd}$ International Conference on Machine Learning*, Seoul, South Korea. PMLR 306, 2026. Copyright 2026 by the author(s).

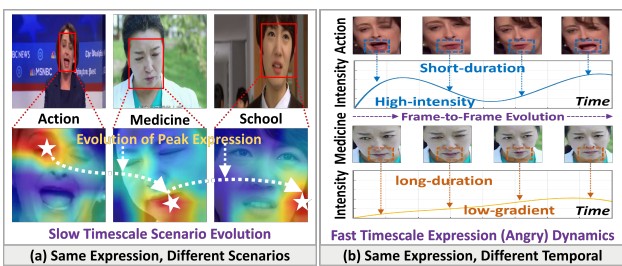

*Figure 1.* Motivation for dual-timescale modeling in DFER under Scenario Variations. Slow scenario evolution (slow timescale) and fast expression dynamics (fast timescale) induce non-stationary distributions of the same expression across scenarios. (a) Scenario-dependent variations in discriminative facial regions. (b) Expression sequences exhibiting dynamic discrepancies across scenarios.

## 1. Introduction

As human-computer interaction (HCI) systems evolve from controlled laboratory environments to real-world physical settings (Slonim et al., 2024; Hosney et al., 2024; Aly, 2025), understanding and predicting human emotions becomes increasingly important for natural and effective interaction. Dynamic Facial Expression Recognition (DFER) (Li & Deng, 2020; Sun et al., 2024; Chen et al., 2024a; Li et al., 2025b) facilitates this by modeling the spatio-temporal evolution of facial expressions in video sequences. However, in long-term real-world deployments, DFER systems inevitably operate under scenario-incremental conditions, where environmental factors such as lighting, background, and viewpoint change over time. Models must therefore continuously process video streams from newly encountered scenarios, often without access to historical raw data due to privacy constraints, storage limitations, or real-time processing requirements (Voigt & Von dem Bussche, 2017; Moore & Frye, 2020). These conditions introduce persistent non-stationarity in observations, posing a fundamental challenge to stable and robust DFER.

Under scenario-incremental deployment, scenario observation conditions (*e.g.*, lighting, background, context, and viewpoint) gradually change over time, distorting the temporal trajectories of facial expressions (Cowen et al., 2021; Wang et al., 2022c), as illustrated in Figure 1. Consequently, the same facial expression can exhibit highly non-stationary appearances across scenarios, making it difficult for existing

DFER methods to accurately model expression dynamics and maintain stable performance over long-term deployment (Chen et al., 2024a; Li et al., 2025b; Chen et al., 2025).

Existing approaches address scenario variations in DFER mainly through two paradigms, both of which have inherent limitations. Cross-domain methods (Li et al., 2022; Wang et al., 2022a; Gao et al., 2024a; Wang & Liu, 2025) aim to reduce distribution shifts between source and target scenarios by aligning observed feature distributions across domains. These approaches typically operate in a passive manner: alignment is performed directly on extracted features after a scenario shift is observed, without explicitly modeling how scenarios evolve over time or maintaining latent states to track such evolution. While effective for reducing static distribution discrepancies, passive feature alignment struggles to capture the continuous temporal dynamics of facial expressions under gradually changing scenarios.

Domain-incremental (no-replay) methods enable sequential learning across multiple scenarios using techniques such as weight regularization (Kirkpatrick et al., 2017), knowledge distillation (Gao et al., 2024b), or architectural expansion (Zhou et al., 2024; Li et al., 2025a; Zhou et al., 2025b) to preserve historical knowledge. These methods effectively mitigate catastrophic forgetting, maintaining long-term knowledge consistency. However, they typically focus on preserving learned representations and rely on a more passive feature discrimination rather than explicitly modeling scenario evolution. As a result, both paradigms rely on passive determination of observed features and lack explicit modeling of scenario evolution, which prevents accurate capture of dynamic facial expressions and weakens long-term stability under continuous, real-world conditions.

To address this limitation, we argue that robust DFER under scenario-incremental deployment requires explicit reasoning about how both scenarios and expressions evolve over time, rather than solely aligning observed features or preserving learned representations. In long-term real-world settings, scenario changes are neither abrupt nor independent; instead, they form a slowly evolving process that continuously influences the appearance and dynamics of expressions. Treating such variations as static domain shifts or as noise inevitably entangles scenario effects with expression dynamics, leading to unstable recognition over time.

Motivated by this observation, we propose **EmWorld**, an emotion world model for DFER with scenario-aware incremental learning. From a world-model perspective (Hao et al., 2023; Zuo et al., 2025; Wang et al., 2024; Ni et al., 2025; Ding et al., 2025; Maes et al., 2026), **EmWorld** formulates scenario-incremental DFER as a progressive Bayesian (Van de Schoot et al., 2021) inference problem over latent world states that evolve at multiple temporal scales. A slow-timescale component (**STS**) explicitly mod-

els scenario evolution as a stochastic process, capturing long-term scenario effects and encoding historical context to proactively guide predictions in new scenarios. A fast-timescale component (**FTS**) models frame-level expression dynamics through temporally consistent latent transitions, enabling stable expression trajectories while decoupling transient expression changes from scenario influences. This unified dual-timescale framework transforms scenario-incremental DFER from passive feature discrimination into active probabilistic inference.

**To summarize, our main contributions are as follows:**

**(1)** We formalize scenario-incremental DFER as progressive Bayesian inference over a latent world state. This framework models scenario evolution at a slow timescale (STS) and expression dynamics at a fast timescale (FTS), shifting recognition from passive feature-based discrimination to a robust active probabilistic reasoning process.

**(2)** We propose **EmWorld**, a dual-timescale model that implements this framework. STS captures long-term scenario progression via stochastic priors, while FTS models frame-level expression dynamics and decouples them from scenario effects, enabling long-term knowledge retention and proactive adaptation to new scenarios.

**(3)** We conduct extensive experiments on FERV39k, DFEW, and MAFW, covering diverse scenarios and task sequences. Results demonstrate that **EmWorld** consistently maintains performance over time, effectively handles new scenarios, and surpasses state-of-the-art methods in both recognition accuracy and cross-scenario stability.

## 2. Related Work

### 2.1. Dynamic Facial Expression Recognition

Dynamic facial expression recognition (DFER) focuses on modeling the temporal evolution of facial expressions from video sequences. Early studies mainly rely on CNNs (Simonyan & Zisserman, 2015; He et al., 2016; Wang et al., 2023), RNNs (Hochreiter & Schmidhuber, 1997), or Transformers (Vaswani et al., 2017; Zhao & Liu, 2021; Li et al., 2023) to capture short-term temporal patterns such as expression intensity variations and phase transitions. Recent advances further explore self-supervised pretraining (Sun et al., 2023; 2024; Chen et al., 2024b; 2025) and multimodal temporal modeling (Li et al., 2025b) to improve representation robustness and reduce annotation costs. Despite these advances, most existing DFER methods assume fixed or single scenarios. In real-world long-term deployment, continuous scenario changes induce continuous distribution shifts, which make it difficult for discriminative models to separate expression dynamics from scenario variations, leading to degraded cross-scenario recognition and reduced

long-term stability under no-replay constraints.

## 2.2. Replay-Free Incremental Learning

Scenario-incremental learning in DFER can be viewed as a domain-incremental setting, where each scenario corresponds to a visual domain. Domain-incremental learning (DIL) addresses settings in which the class set remains fixed while each task introduces a new domain (Wang et al., 2022b; Gao et al., 2024b; Zhou et al., 2025b). Under **replay-free** constraints, weight regularization (Kirkpatrick et al., 2017) penalizes changes to parameters deemed important for previous tasks, helping retain old knowledge; knowledge distillation (Asadi et al., 2023; Bonato et al., 2024; Gao et al., 2024b) transfers predictions from previous models as soft targets to guide learning on new domains; and architectural expansion (Zhou et al., 2024; Li et al., 2025a; Zhou et al., 2025b) adds task-specific modules to accommodate domain shifts. While effective in standard DIL, these methods often treat scenario variations as unstructured noise and rely on passive feature discrimination, failing to capture continuous scenario dynamics and disentangle short-term expression changes from long-term scenario evolution. This can limit their consistency across scenarios and motivates approaches that explicitly model latent state evolution.

## 3. The Proposed Method

### 3.1. Problem Formulation

For scenario-incremental dynamic facial expression recognition (DFER), we focus on the ability of DFER systems to perform continual learning during long-term deployment. Specifically, the model must continuously receive data from new scenarios and learn from them. In this context, the entire data flow can be formally described as:

$$\mathcal{S} = \{\mathcal{S}_k = (\mathcal{X}_k, \mathcal{Z}_k)\}_{k=1}^K, \tag{1}$$

where $\mathcal{X}_k$ and $\mathcal{Z}_k$ denote the training and test sets of the $k$-th scenario, respectively, and $K$ represents the total number of scenarios. Each training set $\mathcal{X}_k = \{(x_k^i, y_k^i)\}_{i=1}^{N_k}$ consists of $N_k$ labeled video sequences, where $\mathbf{x}_k^i$ denotes the $i$-th video clip in the $k$-th scenario, and $\mathbf{y}_k^i \in Y$ is its corresponding expression label. Here, $Y = \{1, 2, 3, \ldots, C\}$ is a fixed label set shared across all scenarios.

During training on scenario $k$, only the current dataset $\mathcal{X}_k$ is accessible, and no historical samples are stored. After training, the model is evaluated on the test sets of all previous scenarios, collectively forming $\mathcal{Z}_{1:k} = \bigcup_{j=1}^k \mathcal{Z}_j$.

### 3.2. Overview

Figure 2 illustrates **EmWorld**, an emotion world model with latent state evolution for scenario-incremental DFER. Motivated by the limitation that traditional approaches passively

rely on observed features, **EmWorld** formulates scenario-incremental DFER as a progressive Bayesian inference problem over a latent world, capturing both slow-timescale (STS) scenario evolution and fast-timescale (FTS) expression dynamics to enable active probabilistic reasoning.

Given an input video sequence $x_{k,1:T}$ from scenario $k$, **EmWorld** maps observations into latent states via a shared encoder and aggregates them into a video-level representation:

$$\mathbf{e}_{k,1:T} = f_\theta(x_{k,1:T}), \quad \mathbf{e}_k = h(\mathbf{e}_{k,1}, \ldots, \mathbf{e}_{k,T}), \tag{2}$$

where $f_\theta(\cdot)$ is a shared encoder (Dosovitskiy et al., 2021), $h(\cdot)$ is the temporal aggregation function, $T$ indexes frames. Although $e_k$ is deterministically derived from the input, **EmWorld** treats it as a sample from a latent world state distribution, capturing uncertainty from gradual scenario drift. Probabilistic modeling is thus performed entirely over latent states and labels, with a unified joint distribution factorized as:

$$P(\mathbf{e}_{k,1:T}, y \mid k) \propto \prod_k P(\boldsymbol{\mu}_{y,k} \mid \boldsymbol{\mu}_{y,k-1}) \prod_t P(\mathbf{e}_{t+1} \mid \mathbf{e}_t, \mathbf{F}_t) P(y \mid \mathbf{e}_k) \tag{3}$$

where $\boldsymbol{\mu}_{y,k}$ denotes the scenario-dependent latent world state associated with expression $y$ at scenario $k$, $F_t$ represents the frame-level latent transition dynamics, and $e_k$ is the aggregated video-level latent representation. This factorization characterizes how latent states and labels are jointly inferred under continuously evolving scenarios.

Each term in Eq. (3) reflects a specific modeling goal in **EmWorld**: the first captures **slow-timescale (STS)** evolution of latent world states across scenarios via stochastic priors, preserving long-term consistency; the transition product models **fast-timescale (FTS)** expression dynamics, decoupling transient variations from scenario-dependent factors. Together, this provides a probabilistic foundation for scenario-incremental DFER, enabling **EmWorld** to retain historical knowledge while maintaining stable long-term performance.

### 3.3. Slow-Timescale Modeling: Scenario Dynamics

To model the long-term impact of scenario changes on latent expression dynamics, we model scenario dynamics at the slow-timescale (STS), focusing on the cross-scenario evolution of latent world states in a statistical scenario. Specifically, the STS mainly consists of two processes: ***Slow-scale Initialization*** and ***Observational Correction***. It models the latent world states corresponding to the same expression across different scenarios as a stochastic process that gradually evolves with the scenario.

***Slow-scale Initialization***: Based on this modeling principle, we assume that in the latent space, the latent world state of a given expression class $c$ under scenario $k$ follows a

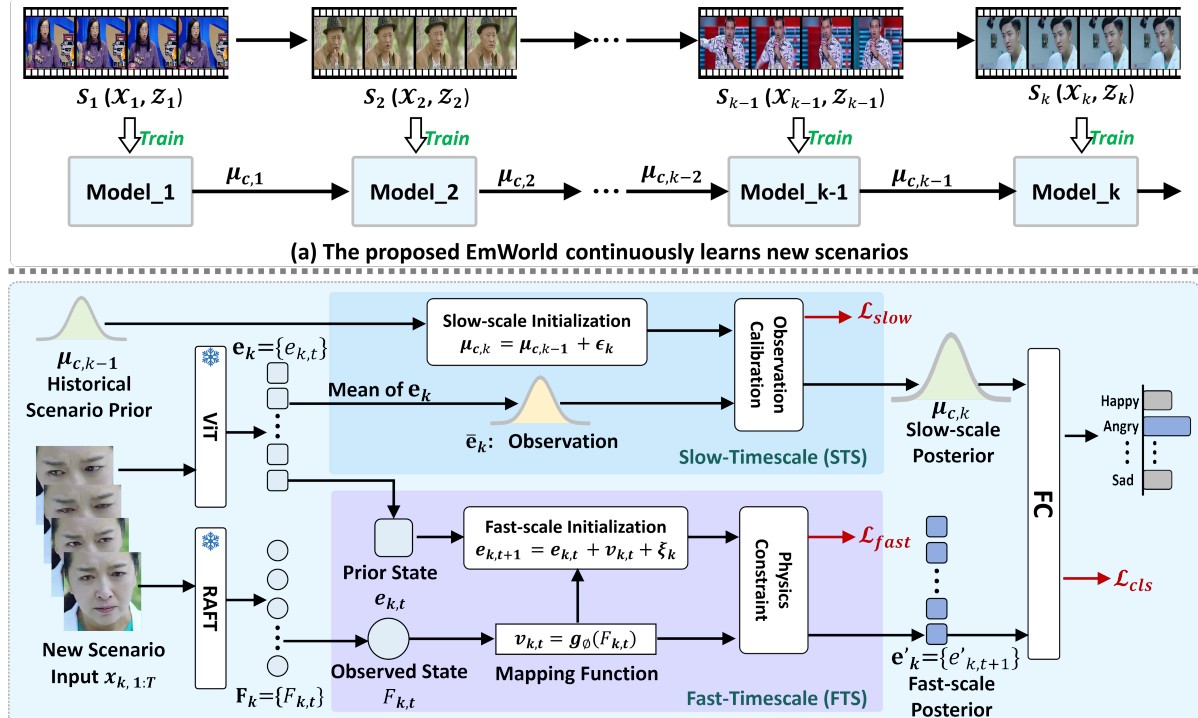

**(a) The proposed EmWorld continuously learns new scenarios**

**(b) Detailed structure of the k-th model in the proposed EmWorld method**

*Figure 2.* The pipeline of ours **EmWorld** for scenario-incremental dynamic facial expression recognition.

class-conditional Gaussian distribution:

$$\mathbf{e}_k \mid (y = c, k) \sim \mathcal{N}(\boldsymbol{\mu}_{c,k}, \boldsymbol{\sigma}_{c,k}), \tag{4}$$

where $\boldsymbol{\mu}_{c,k}$ denotes the center of the latent world state for class $c$ in scenario $k$, capturing systematic shifts induced by the scenario, and $\boldsymbol{\sigma}_{c,k}$ represents the within-class covariance, characterizing transient variations of expressions at a short timescale.

Under this assumption, modeling scenario dynamics naturally reduces to modeling the evolution of the latent world state centers. We adopt a stochastic random walk (West & Harrison, 1997; Durbin & Koopman, 2012) to capture this evolution:

$$\boldsymbol{\mu}_{c,k} = \boldsymbol{\mu}_{c,k-1} + \boldsymbol{\epsilon}_k, \quad \boldsymbol{\epsilon}_k \sim \mathcal{N}(0, \mathbf{Q}), \tag{5}$$

where $\mathbf{Q}$ is the process noise covariance, representing the uncertainty caused by scenario changes. This model acknowledges that the cross-scenario distribution is non-stationary, while imposing a structural smoothness constraint on this non-stationarity: scenario changes do not cause abrupt distribution shifts, but rather a slow, modelable evolution. By modeling scenario dynamics with a slow-timescale random walk, the model can smoothly integrate new scenario information while preserving the integrity of historical latent world states, ensuring long-term recognition stability.

***Observational Correction***: For the above random walk modeling, $\boldsymbol{\mu}_{c,k}$ is an unobserved latent variable that can only be

indirectly estimated from the observed data in the current scenario. Therefore, we further introduce an observation model to characterize the estimation uncertainty arising from the observed data and to combine data-driven information with the dynamic prior of the class center.

Specifically, given the set of samples $\mathcal{S}_{c,k}$ for class $c$ in scenario $k$, the samples $\mathbf{e}_k = \{e_{k,t}\}$ mean is defined as:

$$\bar{\mathbf{e}}_k = \frac{1}{|\mathcal{S}_{c,k}|} \sum_{\mathcal{S}_{c,k}} \mathbf{e}_k, \quad \bar{\mathbf{e}}_k \sim \mathcal{N}\left(\boldsymbol{\mu}_{c,k}, \frac{\theta^2}{|\mathcal{S}_{c,k}|}\mathbf{I}\right). \tag{6}$$

This observation model explicitly captures the uncertainty caused by sample size and statistical fluctuations: when the number of samples is small, the observation noise is large and the current estimate is unreliable; when the sample size is sufficient, the observation becomes more stable. In this way, the uncertainty introduced by scenario changes is no longer ignored but is explicitly quantified and incorporated into the inference process.

After defining the random-walk prior and the observation model, we further perform progressive Bayesian updates of the class center by combining the two. Given historical data $\mathcal{S}_{<k}$, the prior distribution of the class center $\boldsymbol{\mu}_{c,k}$ is:

$$\boldsymbol{\mu}_{c,k} \mid \mathcal{S}_{<k} \sim \mathcal{N}(\boldsymbol{\mu}_{c,k-1}, \boldsymbol{\sigma}_{c,k-1} + \mathbf{Q}) \tag{7}$$

the observation likelihood is:

$$\bar{\mathbf{e}}_k \mid \boldsymbol{\mu}_{c,k} \sim \mathcal{N}\left(\boldsymbol{\mu}_{c,k}, \frac{\theta^2}{|\mathcal{S}_{c,k}|}\mathbf{I}\right). \tag{8}$$

By combining the prior and observation likelihood, we obtain the posterior distribution of the class center:

$$p(\boldsymbol{\mu}_{c,k} \mid \mathcal{S}_{\leq k}) \propto p(\bar{\mathbf{e}}_k \mid \boldsymbol{\mu}_{c,k}) \, p(\boldsymbol{\mu}_{c,k} \mid \mathcal{S}_{<k}), \qquad (9)$$

the maximum a posteriori (MAP) estimate is equivalent to minimizing the following energy function:

$$\mathcal{L}_{\mathrm{p}}(\boldsymbol{\mu}) = (\boldsymbol{\mu} - \boldsymbol{\mu}_{c,k-1})^{\top}(\boldsymbol{\sigma}_{c,k-1} + \mathbf{Q})^{-1}(\boldsymbol{\mu} - \boldsymbol{\mu}_{c,k-1}), \qquad (10)$$

$$\mathcal{L}_{\mathrm{o}}(\boldsymbol{\mu}) = (\bar{\mathbf{e}}_k - \boldsymbol{\mu})^{\top}\left(\tfrac{\theta^2}{|\mathcal{S}_{c,k}|}\mathbf{I}\right)^{-1}(\bar{\mathbf{e}}_k - \boldsymbol{\mu}), \qquad (11)$$

$$\mathcal{L}(\boldsymbol{\mu}) = \mathcal{L}_{\mathrm{p}}(\boldsymbol{\mu}) + \mathcal{L}_{\mathrm{o}}(\boldsymbol{\mu}). \qquad (12)$$

Taking the derivative of $\boldsymbol{\mu}$ in Eq.(12) and setting it to zero yields the Kalman-style update formula (Kalman, 1960; Maybeck, 1982):

$$\mathbf{H}_k = (\boldsymbol{\sigma}_{c,k-1} + \mathbf{Q})\Big((\boldsymbol{\sigma}_{c,k-1} + \mathbf{Q}) + \tfrac{\theta^2}{|\mathcal{S}_{c,k}|}\mathbf{I}\Big)^{-1} \qquad (13)$$

$$\boldsymbol{\mu}_{c,k} = \boldsymbol{\mu}_{c,k-1} + \mathbf{H}_k(\bar{\mathbf{e}}_k - \boldsymbol{\mu}_{c,k-1}) \qquad (14)$$

$$\boldsymbol{\sigma}_{c,k} = (\mathbf{I} - \mathbf{H}_k)(\boldsymbol{\sigma}_{c,k-1} + \mathbf{Q}) \qquad (15)$$

This formulation effectively realizes a weighted fusion of historical knowledge and current observations, allowing the class center to continuously adapt to new scenarios while maintaining long-term consistency and recognition performance across all encountered scenarios.

### 3.4. Fast-Timescale Modeling: Expression Dynamics

To capture the short-term, transient evolution of facial expressions, we model expression dynamics on a fast timescale, focusing on the continuous temporal evolution of latent world states. Specifically, the STS mainly consists of two processes: ***Fast-scale Initialization*** and ***Physical Constraint***. It models the latent representations of consecutive video frames are treated as a smoothly evolving hidden process, with state transitions constrained by the physical motion of facial muscles.

***Fast-scale Initialization***: Given the scenario $k$, we assume that the frame-level latent state $e_{k,t}$ evolves smoothly over time, and that its dynamics can be characterized by a first-order state transition equation (West & Harrison, 1997; Rabiner, 2002):

$$\mathbf{e}_{k,t+1} = \mathbf{e}_{k,t} + v_{k,t} + \xi_t, \quad \xi_t \sim \mathcal{N}(0, \mathbf{R}), \qquad (16)$$

where $v_{k,t}$ denotes the instantaneous change rate of latent features, and $\xi_t$ faollows a Gaussian distribution with covariance matrix $\mathbf{R}$. This formulation reflects the temporal continuity of expressions and provides a unified state-space for imposing physical constraints.

***Physical Constraint***: Relying solely on Eq. (16) may cause $v_{k,t}$ to degenerate into arbitrary latent variations, hindering consistency with actual facial motion. To address this,

we introduce observation-based constraints derived from real physical motion. In videos, expressions are driven by muscle movements and manifest as continuous pixel displacements, which can be captured by optical flow $F_{k,t}$. We first extract video optical flow using the frozen RAFT (Teed & Deng, 2020) and then predict the latent velocity from it via a learnable mapping $g_\phi(\cdot)$.

$$F_{k,t} = \mathrm{RAFT}(x_{k,t}, x_{k,t+1}), \quad v_{k,t} = g_\phi(F_{k,t}), \qquad (17)$$

yielding a frame-level latent transition likelihood:

$$e_{k,t+1} \mid e_{k,t}, F_{k,t} \sim \mathcal{N}\big(e_{k,t} + g_\phi(F_{k,t}), \mathbf{R}\big), \qquad (18)$$

or equivalently, the expected evolution:

$$\mathbb{E}[e_{k,t+1} \mid e_{k,t}, F_{k,t}] = e_{k,t} + g_\phi(F_{k,t}). \qquad (19)$$

This formulation introduces optical flow as an explicit physical observation. By mapping pixel-level motion $F_{k,t}$ to latent-level velocity $v_{k,t}$ via $g_\phi$, the model ensures that latent state transitions are grounded in actual facial muscle dynamics, preventing physically inconsistent jumps that often occur in purely data-driven incremental learning.

### 3.5. Objective Function

For the scenario-incremental DFER task, we adopt the MAP (Burger & Lucka, 2014) strategy during the training of scenario $k$, which combines the data likelihood of the current scenario with the prior from historical scenarios. This approach allows us to impose a smoothness constraint on the latent world state while maintaining classification performance, enabling stable incremental updates. Specifically, the training objective can be written as:

$$\max_{\theta_k} \ \log p(S_k \mid \theta_k) + \log p(\theta_k \mid S_{<k}), \qquad (20)$$

where $\theta_k$ denotes the model parameters learned from the current scenario, $S_k$ represents the data of the current scenario, and $S_{<k}$ denotes the data from historical scenarios. The corresponding loss function can be decomposed into three components: **the classification loss**, **the scenario-consistency loss**, and **the optical-flow constraint loss**.

**The classification loss:** By combining the emotional latent states $\mathbf{e}_k^{'}$ and $\boldsymbol{\mu}_{c,k}$ from the fast- and slow-timescale modules, discriminative learning for known classes is achieved via a cross-entropy loss:

$$\mathcal{L}_{\mathrm{cls}} = -\mathbb{E}_{(x,y)\sim S_k}\left[\log p(y \mid \mathbf{e}_k^{'}, \boldsymbol{\mu}_{c,k})\right], \qquad (21)$$

where $(x, y)$ denotes a training sample and its corresponding label from the current scenario $S_k$.

**The scenario-consistency loss:** During the training of scenario $k$, the Bayesian update in Eq. (4)-Eq. (13) cannot be

directly optimized via backpropagation. Instead, we approximate the MAP inference by marginalizing the latent center $\mu_{c,k}$, leading to a scenario-consistency loss that acts as a soft constraint on the current feature distribution. To address this, we treat the class center $\mu_{c,k}$ as a latent variable and use the mean feature of class $c$ within each batch, $\bar{e}_{c,k}$, as an observation of this latent variable.

Given the historical prior, marginalizing over the latent variable $\mu_{c,k}$ transforms the explicit Bayesian update into a trainable constraint:

$$p(\bar{e}_{c,k} \mid S_{<k}) = \int p(\bar{e}_{c,k} \mid \mu)\, p(\mu \mid S_{<k})\, d\mu, \quad (22)$$

and using the property of Gaussian convolution, we obtain

$$p(\bar{e}_{c,k} \mid S_{<k}) = \mathcal{N}\Big(\mu_{c,k-1},\, \sigma_{c,k-1} + Q + \frac{\theta^2}{|S_{c,k}|}I\Big). \quad (23)$$

Taking the negative log-likelihood and ignoring constant terms yields **the scenario-consistency loss**:

$$O_{c,k} = \sigma_{c,k-1} + Q + \frac{\theta^2}{|S_{c,k}|}I, \quad (24)$$

$$\mathcal{L}_s = (\bar{e}_{c,k} - \mu_{c,k-1})^\top O_{c,k}^{-1}(\bar{e}_{c,k} - \mu_{c,k-1}). \quad (25)$$

During training, $\mu_{c,k-1}$ is frozen historical state. Once the training of scenario $k$ is completed, we perform a one-time discrete update using Eqs. (13)-Eq. (15) to calculate the new prototypes for the subsequent scenario.

This loss encourages the features in the current scenario to remain consistent with the historical world state in the latent space, while using batch-level feature information to guide the latent variable toward the distribution of the new scenario, thereby preserving recognition performance and maintaining long-term stability across scenarios.

**The optical-flow constraint loss:** The optical-flow constraint characterizes facial expression dynamics through the negative log-likelihood of the latent state transitions. Based on the state transition model in Eq. (16), maximizing the frame-level latent state transition probability is equivalent to minimizing its negative log-likelihood (Eq. (18)). Accordingly, the facial expression dynamics constraint can be written as:

$$\mathcal{L}_f = \sum_t \|(e_{k,t+1} - e_{k,t}) - g_\phi(F_{k,t})\|^2 \quad (26)$$

This term can be interpreted as the negative log-likelihood of the latent world model at a fast timescale, constraining the temporal continuity of frame-level latent representations and ensuring that their dynamic changes align with the true facial motion.

By combining the three components, the final joint training objective is formulated as:

$$\mathcal{L} = \mathcal{L}_{\text{cls}} + \lambda_s\, \mathcal{L}_s + \lambda_f\, \mathcal{L}_f, \quad (27)$$

where $\lambda_s$ and $\lambda_f$ control the weights of the scenario-consistency and smooth dynamics constraints, respectively. This design explicitly encourages robust performance maintenance, long-term stability across scenarios, and physically consistent expression modeling, while avoiding explicit reliance on stored historical samples.

## 4. Experiment

To evaluate the robustness of the proposed model **EmWorld**, we conduct experiments on benchmark to compare **EmWorld** to existing state-of-the-art methods. We also provide ablation studies on the components robustness to analyze the effect of different modules.

### 4.1. Experimental Setup

**Dataset:** Following the official benchmark of the **FERV39k** benchmark (Wang et al., 2022c), we evaluate **EmWorld** and existing state-of-the-art methods on its four isolated subsets: Anomaly Issues (**AI9k**), Daily Life (**DL11k**), Strong-Interactive Activities (**SIA10k**), and Weak-Interactive Shows (**WIS9k**). Among these, AI9k comprises 4 scenarios, while DL11k, SIA10k, and WIS9k each contain 6 scenarios. To further verify the robustness of **EmWorld** across different datasets, we also conduct experiments on the **MAFW** (Liu et al., 2022) and **DFEW** (Jiang et al., 2020) datasets. Each dataset comprises 5 sets, with each set treated as an independent scenario to maintain consistency with the scenario-incremental setting. Based on this, for a comprehensive evaluation under the scenario-incremental setup (Lu et al., 2024; Zhou et al., 2025b), we consider *five different scenario orders*. Detailed experimental configurations and performance comparisons for each scenario order are provided in the *Supplementary Material A*.

**Compared Methods:** We compare **EmWorld** against a range of representative methods, including DFER approaches, *i.e.*, MDFER (Sun et al., 2023), SVFAP (Sun et al., 2024), PTH-Net (Li et al., 2025b), and DICL methods, *i.e.*, infLoRA (Liang & Li, 2024), EASE (Zhou et al., 2024), CPrompt (Gao et al., 2024b), DCE (Li et al., 2025a), SimpleCIL (Zhou et al., 2025a), and DUCT (Zhou et al., 2025b). For a fair comparison, all methods employ the same pre-trained backbone.

**Implementation Details:** We conduct all experiments using PyTorch on NVIDIA A100. Following (Wang et al., 2022d; Zhou et al., 2024), we consider the typical pre-trained weights, *i.e.*, ViT-B/16-IN1K (Dosovitskiy et al., 2021), which is pretrained on ImageNet21K (Russakovsky et al., 2015). We optimize **EmWorld** using the AdamW optimizer with a batch size of 32, 15 training epochs, 5 retraining epochs, and a learning rate of 0.001. Moreover, we denote the WAR among all seen scenarios after learning the

*Table 1.* Experiment results on **AI9k**, **DL11k**, **SIA10k**, and **WIS9k** subsets of the FERV39k benchmark. We report average *Avg* and *Last* for different methods among five scenario orders. The best results are indicated in **bold**, and the suboptimal results is underlined.

| Method | AI9k | | DL11k | | SIA10k | | WIS9k | |
|---|---|---|---|---|---|---|---|---|
| | *Avg* | *Last* | *Avg* | *Last* | *Avg* | *Last* | *Avg* | *Last* |
| MDFER (Sun et al., 2023) | 32.00 | 29.49 | 31.32 | 27.45 | 33.64 | 27.94 | 32.73 | 27.68 |
| SVFAP (Sun et al., 2024) | 29.95 | 28.23 | 29.63 | 26.22 | 32.60 | 29.15 | 31.80 | 27.76 |
| PTH-Net (Li et al., 2025b) | 29.94 | 27.59 | 32.95 | 29.79 | 35.44 | 32.71 | 29.45 | 26.03 |
| infLoRA (Liang & Li, 2024) | 14.40 | 6.86 | 11.52 | 4.54 | 12.63 | 4.69 | 12.53 | 5.84 |
| EASE (Zhou et al., 2024) | 12.76 | 8.49 | 13.42 | 8.12 | 13.14 | 7.18 | 12.15 | 6.46 |
| CPrompt (Gao et al., 2024b) | 19.89 | 10.96 | 20.18 | 12.02 | 18.96 | 8.66 | 18.86 | 10.89 |
| DCE (Li et al., 2025a) | 18.63 | 17.90 | 21.66 | 20.40 | 20.61 | 19.12 | 22.48 | 21.58 |
| SimpleCIL (Zhou et al., 2025a) | 20.01 | 14.43 | 20.30 | 15.28 | 24.66 | 19.29 | 23.87 | 18.72 |
| DUCT (Zhou et al., 2025b) | 36.96 | 35.53 | 34.94 | 33.52 | 39.41 | 37.36 | 36.33 | 35.22 |
| **EmWorld** | **40.56** | **39.41** | **38.03** | **34.45** | **42.63** | **40.15** | **40.28** | **38.16** |

*Table 2.* Experiment results on **MAFW** and **DFEW** datasets. We report average *Avg* and *Last* for different methods among five scenario orders.

| Method | DFEW | | MAFW | |
|---|---|---|---|---|
| | *Avg* | *Last* | *Avg* | *Last* |
| infLoRA | 10.60 | 4.67 | 9.95 | 4.79 |
| EASE | 5.18 | 1.74 | 7.17 | 3.84 |
| CPrompt | 20.82 | 9.58 | 21.46 | 12.49 |
| DCE | 47.61 | 49.59 | 39.23 | 39.00 |
| SimpleCIL | 5.02 | 2.26 | 6.50 | 3.23 |
| DUCT | 43.65 | 45.18 | 36.93 | 36.13 |
| **EmWorld** | **53.69** | **54.16** | **42.35** | **41.34** |

*Table 3.* The per-scenario performance comparison on the **AI9k** subset of the FERV39k benchmark. The scenario order (**order1**) during training is *History→Terror→War→Crisis*.

| Method | *History* | *Terror* | *War* | *Crisis* | *Avg* |
|---|---|---|---|---|---|
| MDFER | 36.92 | 29.32 | 32.27 | 29.63 | 32.04 |
| SVFAP | 36.36 | 27.87 | 30.01 | 28.30 | 30.64 |
| PTH-Net | 38.22 | 31.32 | 30.70 | 24.29 | 31.13 |
| infLoRA | 31.54 | 19.88 | 12.86 | 6.55 | 17.71 |
| EASE | 20.41 | 14.98 | 11.69 | 9.97 | 14.26 |
| CPrompt | 32.28 | 20.08 | 14.37 | 10.03 | 19.19 |
| DCE | 21.34 | 18.42 | 20.63 | 17.15 | 19.39 |
| SimpleCIL | 26.53 | 21.23 | 17.47 | 14.47 | 19.93 |
| DUCT | 44.53 | 38.61 | 38.86 | 36.13 | 39.53 |
| **EmWorld** | **48.24** | **38.81** | **40.78** | **38.29** | **41.53** |

$m$-th scenario as $A_m$. For comparison, we mainly consider $A_M$ (**Last**: the performance after learning the last scenario), $\overline{A} = \sum_{m=1}^{M} A_m$ (**Avg**: the average performance among all incremental scenarios), as well as the forgetting measure (FM) (Chaudhry et al., 2018), which quantifies the relative degree of forgetting.

To smooth the integration of historical and new knowledge, we perform incremental weight fusion on the backbone's last layer after each training (Zhou et al., 2025b). The encoder parameters for scenario $k$ are: $\tilde{\theta}_k = \tilde{\theta}_{\text{init}} + \sum_{i=1}^{k} P(\theta_i - \tilde{\theta}_{\text{init}})$, where $\tilde{\theta}_{\text{init}}$ are the backbone's pre-trained parameters, and $P(\cdot)$ scales the current scenario's parameter increments relative to the historical parameters for smooth updates. The classifier is then briefly retrained using $\tilde{\theta}_k$, thereby preserving performance on previous tasks while learning new ones.

### 4.2. Comparison with State-of-the-Arts

Tables 1–2 summarize the performance of different methods on the FERV39k, MAFW and DFEW datasets. Across all datasets (**AI9k**, **DL11k**, **SIA10k**, **WIS9k**, **MAFW**, **DFEW**), our proposed **EmWorld** consistently achieves the highest *Avg* and *Last* accuracies, outperforming previous state-of-the-art methods. All reported results are averaged over five scenario orders, and additional results are provided

in the **Supplementary Material B**.

Specifically, **EmWorld** consistently achieves higher *Avg* than the strongest baseline DUCT, with improvements of 3.60%, 3.09%, 3.22%, 3.95%, 6.08% and 3.12 on the AI9k, DL11k, SIA10k, WIS9k, DFEW and MAFW datasets, respectively. Overall, **EmWorld** achieves an average gain of 3.84% across the six datasets. Moreover, **EmWorld** also achieves superior *Last* performance on all datasets, indicating that the model maintains strong recognition capability even after learning the final scenario.

The per-scenario results (Tables 3 and 4) demonstrate that **EmWorld** maintains strong and consistent performance throughout incremental learning. As illustrated in Figure 3, while some methods mitigate forgetting through conservative updates at the expense of adaptability, **EmWorld** emphasizes effective adaptation to new scenarios, achieving higher accuracy in both early and later stages. For example, on the AI9k subset, **EmWorld** attains an average accuracy of 41.53% across four scenarios, surpassing DUCT (39.53%). On the SIA10k subset, **EmWorld** reaches 48.32% in the first scenario and consistently maintains accuracy above 37% thereafter. Although **EmWorld** exhibits slightly higher forgetting compared to some baselines, its rapid adaptation to new scenarios results in superior overall

*Table 4.* The per-scenario performance comparison on the **SIA10k** subset of the FERV39k benchmark. The scenario order **(order1)** during training is ***Business*→*Experiment*→*Official*→*Crime*→*Interview*→*Contest***.

| Method | *Business* | *Experiment* | *Official* | *Crime* | *Interview* | *Contest* | *Avg* |
|---|---|---|---|---|---|---|---|
| MDFER | 43.58 | 41.26 | 31.35 | 31.33 | 30.01 | 32.68 | 35.05 |
| SVFAP | 40.22 | 40.79 | 30.08 | 34.63 | 27.27 | 32.12 | 34.19 |
| PTH-Net | 36.59 | 43.10 | 32.29 | 33.89 | 30.85 | 36.62 | 35.56 |
| infLoRA | 29.89 | 9.98 | 7.12 | 4.85 | 3.27 | 2.74 | 9.64 |
| EASE | 31.01 | 25.27 | 14.25 | 11.45 | 9.30 | 8.23 | 16.59 |
| CPrompt | 41.90 | 21.87 | 16.23 | 14.36 | 11.67 | 10.38 | 19.41 |
| DCE | 28.49 | 25.05 | 19.79 | 19.82 | 20.14 | 19.20 | 22.08 |
| SimpleCIL | 34.08 | 28.87 | 20.58 | 19.21 | 19.56 | 19.29 | 23.60 |
| DUCT | 39.66 | 38.64 | 36.15 | 34.63 | 35.92 | 36.34 | 36.89 |
| **EmWorld** | **48.32** | **43.52** | **38.79** | **37.89** | **38.36** | **38.83** | **40.95** |

performance.

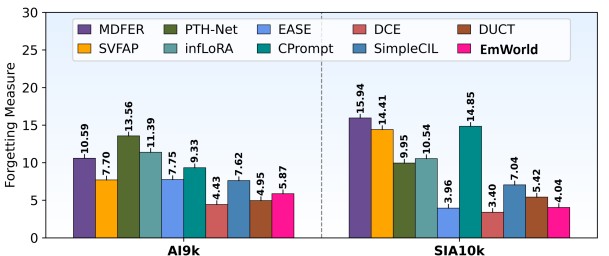

*Figure 3.* Forgetting measure (**lower is better**) of different methods on **AI9k** and **SIA10k** subsets of the FERV39k benchmark with the **order1**

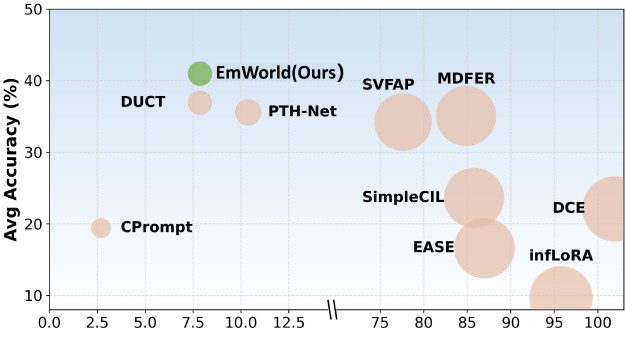

*Figure 4.* Comparison of parameter counts and average accuracy (*Avg*) with state-of-the-art methods on **SIA10k** subset, using the **(order1)** training sequence.

Notably, these improvements are achieved without increased model complexity. As shown in Figure 4, **EmWorld** introduces only lightweight modules and retains a parameter footprint comparable to DUCT, yielding a favorable parameter–performance trade-off. Overall, **EmWorld** achieves a better balance between stability and plasticity, allowing moderate forgetting in exchange for substantially improved per-scenario and overall performance, which is well aligned with the practical requirements of real-world scenario-incremental DFER systems.

### 4.3. Ablation Studies

**Impact of Main Components:** We conduct ablation experiments on the AI9k and SIA10k subsets of the FERV39k

*Table 5.* Ablation results on **AI9k** and **SIA10k** subsets of the FERV39k benchmark. We report average *Avg* and *Last* for different model variants among five scenario orders.

| Variant | STS | FTS | AI9k | | SIA10k | |
|---|---|---|---|---|---|---|
| | | | *Avg* | *Last* | *Avg* | *Last* |
| Baseline | ✗ | ✗ | 37.19 | 35.67 | 39.44 | 37.45 |
| w/ **FTS** | ✗ | ✓ | 37.47 | 36.92 | 40.32 | 38.76 |
| w/ **STS** | ✓ | ✗ | 40.29 | 38.04 | 42.10 | 39.27 |
| **EmWorld** | ✓ | ✓ | **40.56** | **39.41** | **42.63** | **40.15** |

benchmark to analyze the contribution of each component in **EmWorld**. Table 5 reports the *Avg* and *Last* results averaged over five scenario orders. The baseline model achieves relatively limited performance on both datasets, indicating insufficient robustness when learning across evolving scenarios. Introducing the **STS** component leads to clear improvements in *Avg* on both AI9k and SIA10k, demonstrating its effectiveness in maintaining consistent performance across previously observed scenarios by explicitly modeling scenario-level evolution. When further incorporating the **FTS** component, the full **EmWorld** model achieves the best results on both *Avg* and *Last* across datasets. This shows that FTS complements STS by enhancing the modeling of frame-level expression dynamics, which is particularly beneficial for preserving performance after learning later scenarios. Overall, these results confirm that STS and FTS play complementary roles: STS stabilizes representation across scenarios, while FTS improves temporal modeling of expressions, together enabling more stable and reliable scenario-incremental DFER.

**Hyperparameter Sensitivity Analysis:** We conducted sensitivity experiments on the hyperparameters $\lambda_s$, $\lambda_f$, and $Q$ on the AI9k dataset to analyze their impact on performance. The experimental results are shown in Table 6. The results indicate that EmWorld is relatively robust to key hyperparameters. In particular, setting $\lambda_s \approx 0.01$, $\lambda_f \approx 1 \times 10^{-4}$, and $Q \approx 0.1$ provides a good balance between cross-scene stability and frame-level dynamic capture.

**Additional Incremental Learning Metrics:** We include

*Table 6.* Sensitivity analysis of hyperparameters $\lambda_s$, $\lambda_f$, and $Q$ on the AI9k dataset.

| $\lambda_s$ | $\lambda_f$ | $Q$ | *Avg* | *Last* |
|---|---|---|---|---|
| $5 \times 10^{-3}$ | $1 \times 10^{-4}$ | 0.1 | 40.63 | **38.29** |
| 0.02 | $1 \times 10^{-4}$ | 0.1 | 40.75 | 37.89 |
| 0.01 | $5 \times 10^{-5}$ | 0.1 | 40.37 | 37.78 |
| 0.01 | $2 \times 10^{-4}$ | 0.1 | 41.14 | 37.55 |
| 0.01 | $1 \times 10^{-4}$ | 0.05 | 40.06 | 37.04 |
| 0.01 | $1 \times 10^{-4}$ | 0.2 | 39.76 | 37.95 |
| 0.01 | $1 \times 10^{-4}$ | 0.1 | **41.53** | **38.29** |

*Table 7.* Incremental learning performance on the AI9k dataset. Metrics include average *Avg* and *Last*, Forgetting Measure (FM), and Backward Transfer (BWT) among five scenario orders.

| Method | *Avg* | *Last* | *FM* | *BWT* |
|---|---|---|---|---|
| infLoRA | 14.40 | 6.86 | 7.54 | -8.094 |
| EASE | 12.76 | 8.49 | 6.38 | -6.334 |
| CPrompt | 19.99 | 10.96 | 14.14 | -15.092 |
| SimpleCIL | 20.01 | 14.43 | 8.70 | -8.692 |
| DUCT | 37.19 | 35.67 | 3.60 | -4.220 |
| EmWorld | **40.56** | **39.41** | **3.16** | **-3.854** |

commonly used incremental learning metrics from the literature (Chaudhry et al., 2018; Hou et al., 2025): Average Forgetting (FM) and Backward Transfer (BWT). Results on the AI9k dataset are shown in Table 7. The results show that EmWorld outperforms DUCT in both average *Avg* and *Last*, achieving improvements of 3.37% and 3.74%, respectively. Moreover, its forgetting metrics, FM and BWT, also surpass those of the second-best method, DUCT. These findings indicate that the proposed method effectively improves overall recognition performance while maintaining controllable forgetting.

**Computational and Memory Overhead:** The discrete Kalman update is executed once after each scene training stage and is not involved in batch training or inference, resulting in minimal runtime overhead. The update operates on class prototype vectors and their covariance matrices in latent space. In our implementation, a diagonal approximation of the covariance is used, leading to a storage requirement of $O(C \cdot D)$ per class, which remains constant as the number of scenes increases. As shown in Table 8, the Kalman update introduces only a small computational overhead. Across tasks, the runtime increases by 0.75–2.96 s per scene, with an average increase of 1.65 s compared to training without Kalman. Storage is fixed at 10,752 elements for all tasks, corresponding to a buffer maintaining the mean and diagonal covariance vectors for all classes ($2 \times C \times D$, with $C = 7$ and $D = 768$), and does not grow with the number of scenes. Overall, the additional runtime is minor relative to total training time, and the memory requirement remains bounded, indicating that the Kalman update is efficient in both time and space.

**Upper-Bound Experiments:** We report two reference up-

*Table 8.* Runtime overhead of the discrete Kalman update on the AI9k dataset. $\Delta$Time indicates the increase compared to training without Kalman.

| Task | w/o Kalman (s) | w/ Kalman (s) | $\Delta$Time (s) |
|---|---|---|---|
| 0 | 78.21 | 81.17 | +2.96 |
| 1 | 73.20 | 73.95 | +0.75 |
| 2 | 84.87 | 86.01 | +1.14 |
| 3 | 59.97 | 61.71 | +1.74 |
| Avg | 74.06 | 75.71 | +1.65 |

*Table 9.* Performance comparison across scenarios on the AI9k dataset. Single: each scenario trained independently; EmWorld: proposed method; Joint: all scenarios trained together.

| Scene | Single | EmWorld | Joint |
|---|---|---|---|
| History | 48.24 | 48.24 | |
| Terror | 38.39 | 38.81 | 43.82 |
| War | 45.64 | 40.78 | |
| Crisis | 44.85 | 38.29 | |

per bounds: *Single-Scenario Upper Bound*, where each scene is trained independently, and *Joint Training Upper Bound*, where all scenarios are trained together. Results on the AI9k dataset are shown in Table 9. In the initial scenario (History), EmWorld matches single-scenario performance. In subsequent scenarios, it maintains stable performance, with occasional positive transfer. While a gap remains in more challenging later scenarios, EmWorld consistently learns across multiple scenarios.

### 4.4. Limitations

While EmWorld effectively models continuous scenario evolution, its reliance on a Gaussian random walk assumption may lead to a transient adaptation lag during abrupt scenario transitions. A detailed analysis of this trade-off and potential solutions is provided in ***Supplementary Material C***.

### 5. Conclusion and Future Works

In this paper, we propose **EmWorld**, a novel Hierarchical Latent World framework for scenario-incremental dynamic facial expression recognition (DFER) in long-term deployed systems. **EmWorld** explicitly models scenario evolution at the slow timescale (STS) and expression dynamics at the fast timescale (FTS), enabling principled disentanglement of scenario influences from expression trajectories. By combining stochastic evolutionary priors with physically-constrained latent transitions, **EmWorld** mitigates catastrophic forgetting while maintaining stable performance across previously encountered scenarios. For future work, we aim to further reduce forgetting by integrating scenario uncertainty estimation, explore dynamic scenario weighting to handle highly imbalanced scenario sequences, and optimize the latent world update mechanism to mitigate adaptation latency during abrupt transitions, ensuring faster online adaptation in highly dynamic long-term deployments.

## Acknowledgements

This work was supported by the National Natural Science Foundation of China grant (62076227), Natural Science Foundation of Hubei Province grant (2023AFB572) and Hubei Key Laboratory of Intelligent Geo-Information Processing (KLIGIP-2022-B10).

## Impact Statement

This paper presents work whose goal is to advance the field of machine learning. There are many potential societal consequences of our work, none of which we feel must be specifically highlighted here.

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

# A. Dataset Details

This section provides a comprehensive overview of the FERV39k (Wang et al., 2022c), MAFW (Liu et al., 2022), and DFEW (Jiang et al., 2020) datasets used in this study. The FERV39k dataset includes four subsets: AI9k, DL11k, SIA10k, and WIS9k. Specifically, we describe the key characteristics of each dataset, including the total number of scenarios and instances, and detail the five scenario splits used in the main paper for evaluation and analysis.

## A.1. Dataset introduction

**Anomaly Issues (AI9k):** AI9k comprises four rarely encountered scenarios: *History, Terror, War, and Crisis*. These scenarios feature unusual facial appearances and rapid expression changes, posing challenges for both researchers and DFER methods. The seven basic expressions (Angry, Disgust, Fear, Happy, Sad, Surprise, and Neutral) are consistently annotated according to a standardized handbook.

**Daily Life (DL11k):** DL11k contains six everyday scenarios: *Argue, Social, School, Medicine, Conflict, and Daily-Life*. The scenarios differ substantially due to the complexity of real-life activities. To ensure consistent annotation, the seven basic expressions follow the same handbook

**Strong-Interactive Activities (SIA10k):** SIA10k focuses on six strongly interactive activities: *Business, Experiment, Official-Event, Crime, Interview, and Contest*. Emotions in these scenarios are influenced by interactions with others and the environment, resulting in diverse and unstable expression distributions. The seven basic expressions are annotated consistently

**Weak-Interactive Shows (WIS9k):** WIS9k comprises six types of shows: *Action, Scholar Reports, Speech, Elegant-Art, Live-Show, and Talk-Show*. Individuals in these scenarios usually maintain a relatively stable emotional state over longer periods, and the intensity of expressions is generally higher. Annotation follows the same handbook for consistency

**MAFW:** MAFW is divided into five folds, each treated as an independent scenario. The model incrementally learns from the samples in each fold before moving to the next, enabling evaluation across all folds as separate learning stages. Only single-expression clips (11 basic emotions) are used, with annotations including video-level emotion labels and facial landmarks to ensure consistency.

**DFEW:** DFEW is split into five folds, each considered an independent scenario. The model incrementally learns from the samples in each fold before proceeding to the next, allowing evaluation across all folds as separate learning stages. Only single-expression clips (7 basic emotions) are used, with annotations including video-level emotion labels and facial landmarks to ensure consistency.

## A.2. Scenario Sequences

In domain-incremental continual learning, the performance of different algorithms can be influenced by the order in which scenarios are presented. To systematically study this effect, we randomly shuffle the scenarios and define five scenario orders for each subset in the main paper. These orders are used for a holistic evaluation of model robustness. The specific scenario orders for each subset are summarized in Tables 10, 11, 12, 13, and 14:

Table 10 lists the five scenario orders for the four AI9k scenarios (*History, Terror, War, and Crisis*). The different permutations allow us to assess how models cope with rare and unexpected expressions under various sequences.

Table 11 lists the five scenario orders for the six DL11k scenarios (*Argue, Social, School, Medicine, Conflict, and Daily-Life*). By varying the order, we can evaluate whether models can consistently retain knowledge across common daily-life activities.

Table 12 lists the five scenario orders for the six SIA10k scenarios (*Business, Experiment, Official-Event, Crime, Interview, and Contest*). These sequences test the model's ability to handle interactions that introduce diverse and unstable expression distributions.

Table 13 lists the five scenario orders for the six WIS9k scenarios (*Action, Scholar Reports, Speech, Elegant-Art, Live-Show, and Talk-Show*). The varying orders examine the model's performance under prolonged and high-intensity expressions typically seen in show-type scenarios.

Table 14 lists the five scenario orders for the five MAFW scenarios (*set1, set2 set3, set4, set5*). The varying orders examine the model's performance under prolonged and high-intensity expressions typically seen in show-type scenarios.

Table 14 lists the five scenario orders for the five DFEW scenarios (*set1, set2 set3, set4, set5*). The varying orders examine the model's performance under prolonged and high-intensity expressions typically seen in show-type scenarios.

By evaluating models across multiple scenario orders, we can better understand their stability, knowledge retention, and ability to generalize under different domain-incremental conditions. This experimental design provides a comprehensive assessment beyond a single fixed order of tasks.

*Table 10.* Scenario orders for the **AI9k** subset of the FERV39k benchmark.

| AI9k | Order-1 | Order-2 | Order-3 | Order-4 | Order-5 |
|------|---------|---------|---------|---------|---------|
| **Scenario-1(S1)** | History | Terror | War | Crisis | Terror |
| **Scenario-2(S2)** | Terror | War | Crisis | History | History |
| **Scenario-3(S3)** | War | Crisis | History | Terror | Crisis |
| **Scenario-4(S4)** | Crisis | History | Terror | War | War |

*Table 11.* Scenario orders for the **DL11k** subset of the FERV39k benchmark.

| DL11k | Order-1 | Order-2 | Order-3 | Order-4 | Order-5 |
|-------|---------|---------|---------|---------|---------|
| Scenario-1(S1) | Argue | Social | School | Medicine | Conflict |
| Scenario-2(S2) | Social | School | Medicine | Conflict | DailyLife |
| Scenario-3(S3) | School | Medicine | Conflict | DailyLife | Argue |
| Scenario-4(S4) | Medicine | Conflict | DailyLife | Argue | Social |
| Scenario-5(S5) | Conflict | DailyLife | Argue | Social | School |
| Scenario-6(S6) | DailyLife | Argue | Social | School | Medicine |

*Table 12.* Scenario orders for the **SIA10k** subset of the FERV39k benchmark.

| SIA10k | Order-1 | Order-2 | Order-3 | Order-4 | Order-5 |
|--------|---------|---------|---------|---------|---------|
| Scenario-1(S1) | Business | Experiment | Official-Event | Crime | Interview |
| Scenario-2(S2) | Experiment | Official | Crime | Interview | Contest |
| Scenario-3(S3) | Official | Crime | Interview | Contest | Business |
| Scenario-4(S4) | Crime | Interview | Contest | Business | Experiment |
| Scenario-5(S5) | Interview | Contest | Business | Experiment | Official |
| Scenario-6(S6) | Contest | Business | Experiment | Official | Crime |

## B. Supplementary Experimental Results

In this section, we present additional experimental results to further demonstrate the effectiveness of **EmWorld**. Specifically, we provide the detailed analysis of model performance across different scenario sequences.

In the main paper, we conduct experiments on four disjoint subsets of the FERV39k benchmark, as well as MAFW and DFEW, under five scenario orders. Here, we provide detailed results for each order in Tables 15–19 for FERV39k, and Tables 20–21 for MAFW and DFEW. The scenario sequences are described in Section A. Compared with the averaged performance reported in the main paper, these fine-grained results offer a more comprehensive understanding of model behavior under diverse scenario sequences.

We first analyze performance on FERV39k, which consists of four disjoint subsets and thus offers a comprehensive testbed for continual learning. Across these subsets, **EmWorld** consistently outperforms existing methods in most settings. Specifically, among the 40 evaluation metrics (5 scenario orders × 4 subsets × 2 metrics), **EmWorld** achieves the best performance in 38 cases. The only exceptions occur on DL11k: in Order-1, its *Last* score (31.38%) is slightly below PTH-Net (Li et al., 2025b) (31.92%) and DUCT (Zhou et al., 2025b) (33.92%), and in Order-3, its *Last* score (31.07%) is marginally lower than that of DUCT (34.66%). Despite these isolated cases, **EmWorld** maintains clear superiority on the remaining three subsets and consistently achieves the best results across Orders 1, 2, 4, and 5, indicating strong robustness to scenario ordering.

To further evaluate generalization beyond FERV39k, we next consider the MAFW benchmark. On this dataset, **EmWorld** achieves the highest average (*Avg*) and final (*Last*) scores across all five scenario orders, outperforming the second-best method, DCE, by more than 3 percentage points in most cases. For example, in Order-3, **EmWorld** attains an *Avg* of 45.17%

*Table 13.* Scenario orders for the **WIS9k** subset of the FERV39k benchmark.

| WIS9k | Order-1 | Order-2 | Order-3 | Order-4 | Order-5 |
|---|---|---|---|---|---|
| Scenario-1(S1) | Action | ScholarReports | Speech | Elegant-Art | Live-Show |
| Scenario-2(S2) | ScholarReports | Speech | Elegant-Art | Live-Show | Talk-Show |
| Scenario-3(S3) | Speech | Elegant-Art | Live-Show | Talk-Show | Action |
| Scenario-4(S4) | Elegant-Art | Live-Show | Talk-Show | Action | ScholarReports |
| Scenario-5(S5) | Live-Show | Talk-Show | Action | ScholarReports | Speech |
| Scenario-6(S6) | Talk-Show | Action | ScholarReports | Speech | Elegant-Art |

*Table 14.* Scenario orders for the **MAFW** and **DFEW** datasets.

| | MAFW | | | | | DFEW | | | | |
|---|---|---|---|---|---|---|---|---|---|---|
| | Order-1 | Order-2 | Order-3 | Order-4 | Order-5 | Order-1 | Order-2 | Order-3 | Order-4 | Order-5 |
| Scenario-1(S1) | set1 | set2 | set3 | set4 | set5 | set1 | set2 | set3 | set4 | set5 |
| Scenario-2(S2) | set2 | set3 | set4 | set5 | set1 | set2 | set3 | set4 | set5 | set1 |
| Scenario-3(S3) | set3 | set4 | set5 | set1 | set2 | set3 | set4 | set5 | set1 | set2 |
| Scenario-4(S4) | set4 | set5 | set1 | set2 | set3 | set4 | set5 | set1 | set2 | set3 |
| Scenario-5(S5) | set5 | set1 | set2 | set3 | set4 | set5 | set1 | set2 | set3 | set4 |

*Table 15.* The performance comparison on the FERV39k benchmark. We report average and final performance of different methods under **Order-1**. The best and second-best results on each dataset are highlighted in **bold** and underlined, respectively.

| Method | AI9k | | DL11k | | SIA10k | | WIS9k | |
|---|---|---|---|---|---|---|---|---|
| | *Avg* | *Last* | *Avg* | *Last* | *Avg* | *Last* | *Avg* | *Last* |
| MDFER (Sun et al., 2023) | 32.04 | 29.63 | 32.11 | 27.63 | 35.05 | 32.68 | 34.27 | 28.83 |
| SVFAP (Sun et al., 2024) | 30.64 | 28.30 | 30.55 | 25.89 | 34.19 | 32.12 | 33.14 | 28.77 |
| PTH-Net (Li et al., 2025b) | 31.13 | 24.29 | 34.38 | 31.92 | 35.56 | 36.62 | 31.19 | 29.20 |
| infLoRA (Liang & Li, 2024) | 17.71 | 6.55 | 11.32 | 3.37 | 9.64 | 2.74 | 18.65 | 8.26 |
| EASE (Zhou et al., 2024) | 14.26 | 9.97 | 14.74 | 8.94 | 16.59 | 8.23 | 11.80 | 4.47 |
| CPrompt (Gao et al., 2024b) | 19.19 | 10.03 | 23.58 | 11.52 | 19.41 | 10.38 | 20.48 | 10.41 |
| DCE (Li et al., 2025a) | 19.39 | 17.15 | 24.15 | 19.59 | 22.08 | 19.20 | 25.24 | 21.10 |
| SimpleCIL (Zhou et al., 2025a) | 19.93 | 14.47 | 22.34 | 15.25 | 23.60 | 19.29 | 26.54 | 18.72 |
| DUCT (Zhou et al., 2025b) | 39.53 | 36.13 | 39.04 | **33.92** | 36.89 | 36.34 | 37.64 | 34.33 |
| **EmWorld** | **41.53** | **38.29** | **40.67** | 31.38 | **40.95** | **38.83** | **43.04** | **36.82** |

*Table 16.* The performance comparison on the FERV39k benchmark. We report average and final performance of different methods under **Order-2** . The best and second-best results on each dataset are highlighted in **bold** and underlined, respectively.

| Method | AI9k | | DL11k | | SIA10k | | WIS9k | |
|---|---|---|---|---|---|---|---|---|
| | *Avg* | *Last* | *Avg* | *Last* | *Avg* | *Last* | *Avg* | *Last* |
| MDFER (Sun et al., 2023) | 31.17 | 28.55 | 30.72 | 28.72 | 35.98 | 23.80 | 32.29 | 31.40 |
| SVFAP (Sun et al., 2024) | 28.72 | 27.84 | 28.60 | 25.84 | 35.73 | 22.64 | 31.04 | 29.27 |
| PTH-Net (Li et al., 2025b) | 29.28 | 28.50 | 33.01 | 34.57 | 39.09 | 30.85 | 28.43 | 24.75 |
| infLoRA (Liang & Li, 2024) | 13.39 | 7.98 | 9.23 | 1.53 | 13.22 | 2.84 | 15.54 | 6.50 |
| EASE (Zhou et al., 2024) | 10.94 | 8.83 | 11.20 | 7.27 | 12.13 | 5.00 | 17.29 | 7.92 |
| CPrompt (Gao et al., 2024b) | 18.95 | 12.88 | 21.51 | 12.36 | 23.64 | 10.72 | 19.43 | 10.35 |
| DCE (Li et al., 2025a) | 18.47 | 19.26 | 19.92 | 20.33 | 21.74 | 19.49 | 25.22 | 21.38 |
| SimpleCIL (Zhou et al., 2025a) | 19.49 | 14.42 | 21.48 | 15.29 | 26.34 | 19.29 | 24.90 | 18.72 |
| DUCT (Zhou et al., 2025b) | 35.38 | 34.13 | 33.63 | 31.90 | 40.89 | 37.02 | 35.31 | 35.24 |
| **EmWorld** | **38.79** | **39.49** | **40.21** | **37.86** | **45.01** | **39.57** | **39.59** | **38.74** |

and a *Last* of 41.67%, compared to DCE's 41.84% and 39.73%. These consistent gains suggest that the advantages observed on FERV39k extend to a different dataset with distinct data characteristics.

A similar trend is observed on DFEW, which further validates the stability of **EmWorld**. Across all five scenario orders, **EmWorld** achieves the highest values on all 10 evaluation metrics, with particularly notable improvements in Order-2 and

*Table 17.* The performance comparison on the FERV39k benchmark. We report average and final performance of different methods under **Order-3**. The best and second-best results on each dataset are highlighted in **bold** and underlined, respectively.

| Method | AI9k | | DL11k | | SIA10k | | WIS9k | |
|---|---|---|---|---|---|---|---|---|
| | *Avg* | *Last* | *Avg* | *Last* | *Avg* | *Last* | *Avg* | *Last* |
| MDFER (Sun et al., 2023) | 33.69 | 28.06 | 29.80 | 28.83 | 32.19 | 27.94 | 31.86 | 30.12 |
| SVFAP (Sun et al., 2024) | 31.26 | 27.70 | 28.24 | 29.48 | 30.91 | 31.99 | 30.83 | 32.11 |
| PTH-Net (Li et al., 2025b) | 29.77 | 27.65 | 32.05 | 26.68 | 33.20 | 33.47 | 28.97 | 26.97 |
| infLoRA (Liang & Li, 2024) | 12.36 | 4.44 | 8.77 | 4.38 | 12.90 | 5.97 | 7.38 | 4.02 |
| EASE (Zhou et al., 2024) | 14.19 | 8.55 | 11.91 | 8.19 | 15.16 | 9.06 | 10.11 | 6.39 |
| CPrompt (Gao et al., 2024b) | 20.12 | 8.89 | 17.12 | 12.36 | 12.83 | 4.41 | 19.84 | 13.18 |
| DCE (Li et al., 2025a) | 19.01 | 17.61 | 19.67 | 20.55 | 20.08 | 18.95 | 20.06 | 21.83 |
| SimpleCIL (Zhou et al., 2025a) | 20.67 | 14.42 | 19.00 | 15.29 | 25.49 | 19.29 | 21.68 | 18.72 |
| DUCT (Zhou et al., 2025b) | 37.35 | 35.67 | 32.37 | **34.66** | 35.89 | 38.64 | 35.66 | 36.65 |
| **EmWorld** | **42.41** | **40.43** | **34.27** | 31.07 | **39.88** | **41.14** | **38.37** | **38.40** |

*Table 18.* The performance comparison on the FERV39k benchmark. We report average and final performance of different methods under **Order-4**. The best and second-best results on each dataset are highlighted in **bold** and underlined, respectively.

| Method | AI9k | | DL11k | | SIA10k | | WIS9k | |
|---|---|---|---|---|---|---|---|---|
| | *Avg* | *Last* | *Avg* | *Last* | *Avg* | *Last* | *Avg* | *Last* |
| MDFER (Sun et al., 2023) | 32.75 | 30.60 | 31.18 | 24.73 | 31.89 | 26.13 | 33.45 | 27.47 |
| SVFAP (Sun et al., 2024) | 30.35 | 28.65 | 29.73 | 23.65 | 30.16 | 26.90 | 31.87 | 27.91 |
| PTH-Net (Li et al., 2025b) | 29.92 | 28.76 | 32.05 | 29.61 | 32.97 | 30.67 | 29.60 | 22.81 |
| infLoRA (Liang & Li, 2024) | 12.81 | 4.67 | 15.52 | 6.57 | 14.15 | 5.63 | 10.18 | 4.13 |
| EASE (Zhou et al., 2024) | 11.10 | 6.44 | 15.17 | 8.76 | 11.27 | 7.35 | 11.47 | 6.33 |
| CPrompt (Gao et al., 2024b) | 22.82 | 11.68 | 17.60 | 10.91 | 19.22 | 9.5 | 19.44 | 12.27 |
| DCE (Li et al., 2025a) | 18.41 | 17.21 | 21.94 | 20.6 | 19.55 | 19.29 | 21.05 | 21.89 |
| SimpleCIL (Zhou et al., 2025a) | 20.18 | 14.42 | 18.76 | 15.29 | 23.27 | 19.29 | 22.72 | 18.72 |
| DUCT (Zhou et al., 2025b) | 38.09 | 37.09 | 31.93 | 33.48 | 40.40 | 37.27 | 35.60 | 35.01 |
| **EmWorld** | **41.68** | **40.34** | **35.70** | **36.15** | **42.64** | **41.13** | **39.80** | **39.42** |

*Table 19.* The performance comparison on the FERV39k benchmark. We report average and final performance of different methods under **Order-5**. The best and second-best results on each dataset are highlighted in **bold** and underlined, respectively.

| Method | AI9k | | DL11k | | SIA10k | | WIS9k | |
|---|---|---|---|---|---|---|---|---|
| | *Avg* | *Last* | *Avg* | *Last* | *Avg* | *Last* | *Avg* | *Last* |
| MDFER (Sun et al., 2023) | 30.35 | 30.60 | 32.77 | 27.35 | 33.07 | 29.16 | 31.80 | 20.58 |
| SVFAP (Sun et al., 2024) | 28.79 | 28.65 | 31.02 | 26.23 | 31.99 | 32.12 | 32.12 | 20.73 |
| PTH-Net (Li et al., 2025b) | 29.58 | 28.76 | 33.24 | 26.19 | 36.39 | 31.94 | 29.04 | 26.40 |
| infLoRA (Liang & Li, 2024) | 15.75 | 10.66 | 12.76 | 6.84 | 13.24 | 6.27 | 10.90 | 6.28 |
| EASE (Zhou et al., 2024) | 13.31 | 8.66 | 14.09 | 7.45 | 10.54 | 6.27 | 10.08 | 7.18 |
| CPrompt (Gao et al., 2024b) | 18.37 | 11.34 | 21.10 | 12.97 | 19.72 | 8.28 | 15.13 | 8.26 |
| DCE (Li et al., 2025a) | 17.89 | 18.29 | 22.62 | 20.95 | 19.60 | 18.66 | 20.84 | 21.72 |
| SimpleCIL (Zhou et al., 2025a) | 19.76 | 14.42 | 19.92 | 15.29 | 24.59 | 19.29 | 23.52 | 18.72 |
| DUCT (Zhou et al., 2025b) | 35.60 | 35.33 | 37.93 | 34.31 | 43.14 | 38.00 | 37.11 | 34.73 |
| **EmWorld** | **38.40** | **38.52** | **39.30** | **35.80** | **44.68** | **40.06** | **40.58** | **37.44** |

Order-5. For instance, in Order-2, **EmWorld** reaches a *Last* score of 55.35%, exceeding DCE by more than 5%. Even under highly variable scenario sequences, **EmWorld** maintains low forgetting, highlighting its robustness against catastrophic forgetting.

Overall, by jointly considering results on FERV39k, MAFW, and DFEW, the above analyses consistently demonstrate that **EmWorld** exhibits strong adaptability and stable performance under complex and dynamic scenario sequences. These findings corroborate its effectiveness as a robust continual learning approach.

*Table 20.* The performance comparison on the MAFW dataset. We report average and final performance of different methods under **Order-1** to **Order-5**. The best and second-best results on each dataset are highlighted in **bold** and underlined, respectively.

| Method | Order-1 | | Order-2 | | Order-3 | | Order-4 | | Order-5 | |
|---|---|---|---|---|---|---|---|---|---|---|
| | *Avg* | *Last* | *Avg* | *Last* | *Avg* | *Last* | *Avg* | *Last* | *Avg* | *Last* |
| infLoRA (Liang & Li, 2024) | 9.30 | 4.80 | 8.82 | 4.69 | 9.64 | 4.69 | 10.74 | 4.91 | 11.26 | 4.85 |
| EASE (Zhou et al., 2024) | 4.50 | 3.44 | 6.70 | 4.10 | 7.36 | 3.88 | 8.51 | 3.61 | 8.77 | 4.15 |
| CPrompt (Gao et al., 2024b) | 17.45 | 12.35 | 22.58 | 12.94 | 23.08 | 11.91 | 22.98 | 13.10 | 21.20 | 12.13 |
| DCE (Li et al., 2025a) | 35.90 | 39.51 | 41.98 | 39.30 | 41.84 | 39.73 | 40.52 | 38.60 | 35.91 | 37.84 |
| SimpleCIL (Zhou et al., 2025a) | 5.05 | 3.23 | 5.92 | 3.23 | 6.36 | 3.23 | 7.65 | 3.23 | 7.50 | 3.23 |
| DUCT (Zhou et al., 2025b) | 33.66 | 36.98 | 38.80 | 36.23 | 38.54 | 36.39 | 38.54 | 35.69 | 35.09 | 35.36 |
| **EmWorld** | **39.52** | **41.29** | **43.72** | **41.13** | **45.17** | **41.67** | **43.63** | **41.35** | **39.70** | **41.24** |

*Table 21.* The performance comparison on the DFEW dataset. We report average and final performance of different methods under **Order-1** to **Order-5**. The best and second-best results on each dataset are highlighted in **bold** and underlined, respectively.

| Method | Order-1 | | Order-2 | | Order-3 | | Order-4 | | Order-5 | |
|---|---|---|---|---|---|---|---|---|---|---|
| | *Avg* | *Last* | *Avg* | *Last* | *Avg* | *Last* | *Avg* | *Last* | *Avg* | *Last* |
| infLoRA (Liang & Li, 2024) | 10.70 | 4.60 | 10.60 | 5.28 | 10.62 | 4.13 | 10.93 | 3.92 | 10.15 | 5.41 |
| EASE (Zhou et al., 2024) | 5.07 | 1.41 | 6.44 | 1.53 | 4.12 | 1.75 | 5.83 | 2.05 | 4.43 | 1.96 |
| CPrompt (Gao et al., 2024b) | 20.09 | 9.16 | 20.41 | 9.16 | 21.62 | 9.93 | 20.52 | 8.90 | 21.48 | 10.74 |
| DCE (Li et al., 2025a) | 46.04 | 49.21 | 46.14 | 49.51 | 49.15 | 49.72 | 49.45 | 50.23 | 47.29 | 49.30 |
| SimpleCIL (Zhou et al., 2025a) | 5.46 | 2.26 | 4.97 | 2.26 | 4.30 | 2.26 | 5.63 | 2.26 | 4.74 | 2.26 |
| DUCT (Zhou et al., 2025b) | 43.10 | 44.44 | 42.68 | 45.68 | 44.17 | 45.12 | 43.73 | 45.29 | 44.55 | 45.38 |
| **EmWorld** | **52.86** | **53.56** | **53.80** | **55.35** | **54.66** | **53.73** | **52.94** | **54.15** | **54.20** | **54.03** |

## C. Discussion and Limitations

While EmWorld provides a robust and principled framework for scenario-incremental DFER, it is important to discuss the underlying assumptions of the proposed Slow-Time Scale (STS) module, which represent a deliberate design choice tailored to long-term and gradually evolving deployment scenarios. Specifically, the STS component models scenario evolution as a *Gaussian random walk* in the latent space. This assumption effectively captures smooth and continuous environmental variations, such as gradual illumination changes, progressive background shifts, or slowly varying contextual factors, which are commonly encountered in long-term real-world deployments.

However, in certain extreme or highly volatile real-world applications, scenario transitions may be abrupt or discontinuous rather than smoothly evolving. For example, a sudden camera hand-off from an indoor environment to a high-contrast outdoor scene, or an instantaneous change in camera viewpoint, can induce a step-change in the underlying scenario distribution. Under such conditions, the Gaussian random walk prior in the STS module may exhibit a brief adaptation lag, as the Bayesian posterior requires multiple observations to recalibrate to the new scenario. During this transition phase, the decoupling between scenario effects and expression dynamics may be temporarily less accurate.

It is worth noting that this behavior reflects an inherent trade-off between stability and responsiveness: the STS module prioritizes long-term consistency and robustness across scenarios, which is critical for replay-free continual learning, at the cost of slower adaptation to abrupt distribution shifts. Addressing this limitation offers a promising direction for future research. In particular, integrating **stochastic jump processes**, **change-point detection**, or hybrid smooth–discrete latent dynamics into the world model could enable the latent state to adaptively "reset" or "jump" when a significant scenario shift is detected. Such extensions would further enhance the agility of EmWorld in highly dynamic environments, while preserving its long-term stability guarantees, and we leave this exploration as an interesting avenue for future work.

