# OpenReview forum: "EmWorld: Emotion World Model with Latent State Evolution for Scenario-Incremental Dynamic Facial Expression Recognition"
_ICML.cc/2026/Conference — ICML 2026 regular_

### Official Review · Reviewer_ASz3 · 2026-03-08

**Soundness:** 3
**Presentation:** 3
**Significance:** 3
**Originality:** 3
**Overall Recommendation:** 4
**Confidence:** 3

**Summary:**

This study focuses on  the task of scenario-incremental dynamic facial expression recognition. The authors argue that existing methods mainly focus on feature alignment or knowledge preservation, while lacking explicit modeling of both scenario evolution over time and short-term expression dynamics. To address this issue, they propose EmWorld, a dual-timescale framework designed for this setting. Experiments are conducted on FERV39k, DFEW, and MAFW, and the results show that EmWorld consistently outperforms prior methods in terms of Avg and Last accuracy.

**Compliance With Llm Reviewing Policy:**

Affirmed.

**Final Justification:**

My concerns have been addressed.

**Key Questions For Authors:**

Please see the weaknesses.

**Limitations:**

Yes

**Strengths And Weaknesses:**

Strengths:
1. The overall idea of this work is clear. It adopts a dual-timescale modeling strategy to separately handle long-term evolution across scenarios and short-term expression dynamics within videos. This design is well aligned with the task motivation and appears reasonable. In particular, the STS part is relatively well developed. The paper provides a complete formulation with class-conditional Gaussian modeling, a random-walk prior, an observation model, and a Kalman-style update process. Although the overall system is still mainly based on discriminative learning in the end, the STS module at least has a relatively solid statistical interpretation, which strengthens the technical credibility of the method.
2. On the experimental side, the authors evaluate the method on FERV39k, DFEW, and MAFW, and report results averaged over multiple scenario orders. The results show that the method consistently outperforms the compared approaches on both Avg and Last metrics. Overall, the empirical results are stable and provide good support for the effectiveness of the method under this task setting.

Weaknesses:
1. The boundary between Contribution 1 and Contribution 2 is somewhat unclear, as both emphasize the importance of STS and FTS, and their distinction is not sufficiently strong. From the method details and the ablation results, the main gains of the paper come from STS. In contrast, FTS is more of a short-term dynamic constraint based on optical flow. While its design is reasonable, it is weaker than STS in terms of both theoretical depth and originality, and it appears more like a task-specific auxiliary module.
2. Both the motivation and the method description repeatedly emphasize the “disentanglement” between scenario variation and expression dynamics. However, the current evidence still mainly comes from final performance improvements and module ablations. More direct representation-level analysis, such as latent space visualization or more targeted factor-analysis experiments, is missing. Therefore, the current experiments mainly demonstrate that the method is effective, but they do not yet fully establish that it achieves the claimed structural disentanglement.

---

> ### Author Rebuttal · Authors · 2026-03-31
>
> We sincerely thank the reviewers for their constructive comments. Based on these suggestions, we provide additional analyses and experiments to address the two questions you raised. A detailed response is summarized as follows:
> # Q1: Clarification of Contribution Boundaries
> We clarify the distinction and relationship between STS and FTS. STS primarily addresses scenario-incremental changes by modeling long-term distribution evolution, leading to significant performance gains across scenarios. In contrast, FTS focuses on short-term dynamics, enforcing temporal consistency and physical plausibility of expression trajectories through optical flow constraints. These modules operate on different temporal scales: STS captures long-term scene evolution, while FTS ensures that each expression segment is interpreted correctly. Notably, STS provides deeper theoretical contributions by modeling the evolution of scenario distributions, while FTS plays a complementary role in enhancing temporal coherence and dynamic modeling accuracy. In ablation studies on the AI9k dataset, STS alone increases the cross-scenario average accuracy by 3.10%, whereas FTS alone contributes 0.28%. The relatively small gain from FTS is due to the primary bottleneck arising from cross-scenario distribution shifts. Nevertheless, FTS is crucial for improving dynamic modeling capability and interpretability, and when combined with STS, it helps maintain both robust performance and temporally coherent predictions. In summary, STS determines “how far” the model can generalize across scenarios, while FTS ensures “how accurately” each segment is modeled, jointly guaranteeing both performance and interpretability.
> # Q2: Empirical Analysis of Decoupling
> We provide preliminary evidence for structured separation of scene and frame-level dynamics. First, as shown in Eq. 3 of the main paper, the joint distribution is explicitly decomposed into STS and FTS. STS updates the scene state, while FTS models frame-level temporal transitions, probabilistically separating cross-scenario evolution from expression dynamics. Empirically, on the AI9k dataset, intra-class variance across scenarios is analyzed as follows:
> |Method|Baseline|Baseline(w/ STS)|EmWorld|
> |-|-|-|-|
> | variance |2.73e-4|0.8e-4|0.9e-4 |
>
> Introducing STS reduces cross-scenario variance by approximately 70%, indicating that scene-related interference is effectively suppressed and expression representations become more stable across different scenarios. On top of this stabilized representation, FTS models frame-level temporal transitions without altering the scene-level state. These results provide preliminary empirical support that the proposed STS and FTS modules enable a structured separation between scene variations and expression dynamics. Additional visual analyses will be included in the final version to further illustrate this decoupling effect.

---

> > ### Author Rebuttal · Reviewer_ASz3 · 2026-04-02
> >
> > My concerns have been addressed. I will keep my score.

---

> > > ### Author Response · Authors · 2026-04-05
> > >
> > > We sincerely thank you for your positive feedback and for confirming that the concerns have been fully resolved. We appreciate your time and support during the review. All clarifications and additional details provided in our response will be faithfully integrated into the final manuscript.

---

### Official Review · Reviewer_jwQX · 2026-03-12

**Soundness:** 4
**Presentation:** 3
**Significance:** 3
**Originality:** 3
**Overall Recommendation:** 5
**Confidence:** 3

**Summary:**

This paper proposes EmWorld, an emotion world model for scenario-incremental Dynamic Facial Expression Recognition. The authors formulate scenario-incremental DFER as a progressive Bayesian inference problem and introduce a dual-timescale inference framework: a Slow Timescale module to model scenario evolution and a Fast Timescale module to capture frame-level expression dynamics. By jointly inferring latent states across these two timescales, the method aims to decouple scenario variations from expression dynamics and improve robustness under evolving deployment conditions.

**Compliance With Llm Reviewing Policy:**

Affirmed.

**Final Justification:**

The authors' rebuttal effectively resolved my initial concerns regarding the forgetting trade-off, base recognition capability, and computational overhead with convincing empirical evidence. I raise my score and request that the authors incorporate these supplementary experiments into the revised manuscript.

**Key Questions For Authors:**

1. Is the reported accuracy WAR or UAR? It would also be helpful to report joint all scenario training Upper Bound and single scenario Upper Bound for clearer performance reference.

2. On at least one dataset, could the authors compare single-scenario recognition performance of EmWorld with standard SOTA methods under WAR/UAR metrics, to verify that the base recognition capability is not significantly compromised by the incremental framework?

3. Please provide ablation studies or additional clarification for the hyperparameters λ_s, λ_f, Q.

4. Since a discrete Kalman update is performed after each scenario, how does the computational and memory cost (e.g., prototype storage and covariance matrix maintenance) scale as the number of scenarios increases?

**Limitations:**

yes

**Strengths And Weaknesses:**

Strengths

- The paper considers realistic deployment settings where factors such as illumination, background, and viewpoint change over time. Modeling scenario-incremental DFER as progressive Bayesian inference offers a novel perspective on this problem.
- The experimental evaluation is comprehensive, including comparisons with both DFER-specific methods and mainstream continual learning approaches.
- The paper is clearly written and well structured, making the methodology easy to follow.



Weaknesses

- Figure 3 shows that the Forgetting Measure of EmWorld is higher than DUCT and some baselines in several cases. The paper explains this as trading moderate forgetting for higher overall accuracy, but the discussion lacks a deeper analysis of whether such a trade-off is appropriate across different application scenarios.
- The paper provides limited analysis of hyperparameter sensitivity.

---

> ### Author Rebuttal · Authors · 2026-03-31
>
> We sincerely thank the reviewer for the constructive comments. A detailed summary of our responses is as follows.
> # Q1: Analysis of Forgetting Trade-off
> The slightly higher forgetting (FM) in Figure 3 was based on a single incremental sequence (order-1), which can exaggerate forgetting. Averaged over five incremental orders on AI9k, the results are:
> |Method|Avg|Last|FM|BWT|
> |-|-|-|-|-|
> |infLoRA|14.40|6.86|7.54|-8.094|
> |EASE|12.76|8.49|6.38|-6.334|
> |CPrompt|19.99|10.96|14.14|-15.092|
> |SimpleCIL|20.01|14.43|8.70|-8.692|
> |DUCT (base)|37.19|35.67|3.60|-4.220|
> |EmWorld|**40.56**|**39.41**|**3.16**|**-3.854**|
>
> EmWorld outperforms DUCT with +3.37 Avg, +3.74 Last WAR, reduces FM by 0.44, and improves BWT by 0.366. This shows that EmWorld effectively retains prior knowledge while adapting to new scenes. This enables robust performance across diverse incremental DFER scenarios, and the trade-off observed in early experiments appears limited to specific sequences, with averaged results providing a more reliable assessment.
>
> # Q2: Hyperparameter Sensitivity Analysis
> We conducted sensitivity experiments on the hyperparameters λ_s, λ_f, and Q on the AI9k dataset to analyze their impact on performance. The experimental results are as follows:
> |λ_s| λ_f|Q|Avg|Last|
> |-|-|-|-|-|
> |5e-3|1e-4|0.1|40.63|38.29|
> |0.02|1e-4|0.1|40.75|37.89|
> |0.01|5e-5|0.1|40.37|37.78|
> |0.01|2e-4|0.1|41.14|37.55|
> |0.01|1e-4|0.05|40.06|37.04|
> |0.01|1e-4|0.2|39.76|37.95|
> |0.01|1e-4|0.1|**41.53**|**38.29**|
>
> EmWorld is robust to hyperparameters, achieving its best Avg/Last WAR of 41.53/38.29 on AI9k with λ_s≈0.01, λ_f≈1e-4, Q≈0.1, balancing cross-scene stability and frame-level dynamics.
> # Q3: Metric Description and Upper-Bound Experiments
> All accuracy results reported in the paper use WAR to ensure fair comparison with existing methods (e.g., DUCT). We further provide two reference upper bounds: Joint Training Upper Bound, where data from all scenes are trained together, and Single-Scene Upper Bound, where each scene is trained and tested independently. The experimental results on the AI9k dataset are as follows:
> |Scene|Single|EmWorld|Joint|
> |-|-|-|-|
> |History|48.24|48.24|43.82|
> |Terror|38.39|38.81||
> |War|45.64|40.78||
> |Crisis|44.85|38.29||
>
> The results show that in the initial scene, EmWorld achieves performance comparable to single-scene training. In subsequent scenes, it maintains stable performance, with positive transfer observed in some cases. Although a gap remains in more challenging later scenes, the model still demonstrates stable performance while continuously learning across multiple scenes.
> # Q4: Verification of Base Recognition Ability
> In our setting, the training process in Scene 1 is equivalent to standard single-scene training. The experimental results on the AI9k dataset are shown below.
> |Method|History|Terror|War|Crisis|Avg|
> |-|-|-|-|-|-|
> |MDFER|36.92|29.32|32.27|29.63|32.04|
> |SVFAP|36.36|27.87|30.01|28.30|30.64|
> |PTH-Net|38.22|31.32|30.70|24.29|31.13|
> |infLoRA|31.54|19.88|12.86|6.55|17.71|
> |EASE|20.41|14.98|11.69|9.97|14.26|
> |CPrompt|32.28|20.08|14.37|10.03|19.19|
> |DCE|21.34|18.42|20.63|17.15|19.39|
> |SimpleCIL|26.53|21.23|17.47|14.47|19.93|
> |DUCT|44.53|38.61|38.86|36.13|39.53|
> |EmWorld|**48.24**|**38.81**|**40.78**|**38.29**|**41.53**|
>
> EmWorld achieves 48.24% WAR in Scene 1. outperforming current non-incremental SOTA methods such as SVFAP and PTH-Net by 11.88 and 10.02, respectively.. This indicates that introducing incremental learning components does not significantly weaken the basic recognition capability of EmWorld.
> # Q5: Computational and Memory Overhead
> The discrete Kalman update is executed once after each scene training stage and is not involved in batch training or inference, resulting in minimal runtime overhead. The update operates on class prototype vectors and their covariance matrices in latent space. In our implementation, a diagonal approximation of the covariance is used, leading to a storage requirement of O(C·D) per class, which remains constant as the number of scenes increases. Experimental results on the AI9k dataset are shown below:
> |Task| w/o Kalman|w/ Kalman| ΔTime | Storage Increase|
> |-|-|-|-|-|
> |0|78.211s|81.171s|+2.961s|10752|
> |1|73.200s|73.950s|+0.750s|10752|
> |2|84.873s|86.009s|+1.136s|10752|
> |3|59.967s|61.706s|+1.739s|10752|
> |Average|74.063s| 75.709s|+1.647s|10752|
>
> The Kalman update introduces only a small computational and memory overhead. Across tasks, the runtime increases by 0.75–2.96 s per scene, with an average increase of 1.65 s compared to training without Kalman. Storage is fixed at 10,752 elements for all tasks, corresponding to a buffer maintaining the mean and diagonal covariance vectors for all classes (2 × C × D, with C = 7 and D = 768), and does not grow with the number of scenes. Overall, the additional runtime is minor relative to total training time, and the memory requirement remains bounded, indicating that the Kalman update is efficient in both time and space.

---

> > ### Author Rebuttal · Reviewer_jwQX · 2026-04-02
> >
> > Thank you for the detailed rebuttal and additional experiments, which effectively resolve my concerns. Please incorporate these new results into the revised manuscript.

---

> > > ### Author Response · Authors · 2026-04-05
> > >
> > > We sincerely thank you for your positive assessment and for confirming that our rebuttal and additional experiments resolved your concerns. As you requested, we will faithfully incorporate the new results and clarifications into the final version of the manuscript. We appreciate your time and supportive guidance.

---

### Official Review · Reviewer_GCw2 · 2026-03-12

**Soundness:** 2
**Presentation:** 3
**Significance:** 2
**Originality:** 2
**Overall Recommendation:** 3
**Confidence:** 5

**Summary:**

The authors propose a Hierarchical Latent World framework, termed EmWorld, for scenario-incremental dynamic facial expression recognition (DFER) in long-term deployed systems. EmWorld explicitly models scenario evolution at the slow timescale and expression dynamics at the fast timescale, enabling principled disentanglement of scenario influences from expression trajectories. EmWorld mitigates catastrophic forgetting while maintaining stable performance across previously encountered scenarios.

**Compliance With Llm Reviewing Policy:**

Affirmed.

**Final Justification:**

I thank the authors for the two rebuttals and for providing additional analyses. The clarifications are helpful and improve the presentation of the work.

The paper addresses an interesting and relevant problem in scenario-incremental DFER and the empirical results are promising overall. However, I still have enough concern about the paper's positioning and empirical grounding that I cannot support acceptance confidently. For these reasons, and considering the positive assessments from the other reviewers, I am increasing my score to Weak Reject.

My main reservations are still only partially resolved. In particular, I remain unconvinced that the current method fully justifies the stronger “emotion world model” framing, as the implementation and evaluation support it more clearly as a structured latent dynamical framework with slow/fast timescale regularization than as a world model in the stronger contemporary sense. I also believe the evidence for separation of scenario and expression dynamics remains indirect, and the use of folds as proxy scenarios on MAFW/DFEW is a practical but imperfect test of the paper’s central hypothesis. The added analyses help, but these issues are tied to the core framing and validation of the work rather than minor presentation details.

**Key Questions For Authors:**

1) Why is the term “world model” appropriate here, beyond latent temporal regularization? What properties of world models are actually demonstrated experimentally?

2) On MAFW/DFEW datasets, why is treating folds as scenarios a faithful test of scenario evolution?

3) Which baselines have been designed for the generic scenario-incremental setting, which ones have been designed for affective computing/emotion recognition scenario-incremental setting, which are sota and video-based DFER? Have the authors re-implemented any of these methods?

**Limitations:**

Limitations are adequately discussed.

There is no discussion at all regarding potential negative societal impact of the work.

**Strengths And Weaknesses:**

++ The paper is easy to read and follow

++ The proposed approach is reasonable and interesting.

-- Throughout the whole paper, authors mention present and mention many times about their developed 'emotion world model'. However, this is over claimed as the method does not appear to learn an explicit environment model in the usual sense, as is typical with current world models. The proposed method is like a structured latent regularization framework with two components (one capturing slow scenario changes; another captuing fast expression dynamics), rather than a world model.

-- To add to my previous weakness, the proposed method's novelty is somewhat limited and incremental, as its components (continual learning, latent temporal modeling, Gaussian evolution priors, temporal consistency constraints) are already known and well-established.

-- There is no direct analysis that support the claim-interpretation that the two method's components actually separate scenario information from expression dynamics.

-- Regarding experiments on MAFW and DFEW datasets: the authors treat each fold as a separate scenario. It does not seem logical/certain that these partitions correspond to meaningful changes in visual context and/or environment and thus the evaluation may not fully test the main hypothesis about modeling gradual scenario evolution. Moreover, since sota methods were not designed for this altered formulation, the comparison with them may not be entirely fair.

-- The ablation study is not fully informative. It mainly compares the full model to variants without the slow/fast components. Other aspects of the design (e.g., choice of Gaussian random walk prior, sensitivity to hyperparameters, or comparisons with simpler alternatives) are not explored nor presented. This is essential for a reader/reviewer to see which parts of the model are actually responsible for the improvements.

-- In terms of catastrophic forgetting, additional continual-learning measures (e.g., backward transfer, per-task accuracy trajectories) will be needed to be added. Additionally, as the authors note, EmWorld exhibits slightly higher forgetting compared to some baselines. This observation raises questions about the author's claim that the proposed approach mitigates catastrophic forgetting.

-- Last but not least, it is not clear which baselines have been designed for the generic scenario-incremental setting, which ones have been designed for affective computing/emotion recognition scenario-incremental setting, which are sota etc. It is thus difficult to interpret how appropriate each baseline is for the evaluated problem. Some stronger video-based DFER models are needed for comparison purposes and/or some more carefully adapted baselines.

-- To the best of my knowledge, both MAFW and DFEW provide video-level emotion annotations rather than frame-level labels (that the authors mention).

---

> ### Author Rebuttal · Authors · 2026-03-31
>
> We thank the reviewer for the constructive comments. Our responses follow:
> # Q1: Definition of “Emotional World Model”
> We clarify that EmWorld goes beyond latent regularization by explicitly modeling latent emotional state evolution with transition dynamics, consistent with modern latent world-model formulations[1–3]. These studies explore latent predictive models of state evolution learned in representation spaces, rather than explicitly reconstructing the environment. Specifically, we formulate scenario-incremental DFER as progressive Bayesian inference over latent world states, STS models stochastic scenario evolution, and FTS models temporal expression dynamics. Together, they form a dual-timescale latent transition model capturing state–transition–prediction structure. EmWorld follows a similar latent predictive formulation by modeling emotional state evolution under scenario drift. This formulation suits non-stationary DFER, where evolving environmental conditions influence expression trajectories. Unlike passive feature alignment, EmWorld explicitly models scenario evolution for predictive reasoning.
>
> [1] A Path Towards Autonomous Machine Intelligence
>
> [2] V-JEPA: Video Joint Embedding Predictive Architecture
>
> [3] LeWorldModel: Stable End-to-End Joint-Embedding Predictive Architecture from Pixels
> # Q2: Method Novelty
> The novelty of EmWorld lies in a world-model-based formulation for scenario-incremental DFER, where learning is modeled as two interacting temporal processes: slow scenario evolution and fast expression dynamics. Environmental factors change gradually across scenarios, while facial expressions evolve rapidly within videos. Existing approaches treat scenario variation as domain shift or forgetting, often entangling environment drift with expression dynamics. EmWorld instead formulates the problem as dual-timescale stochastic latent-state evolution: STS models cross-scenario drift of latent emotional states, while FTS models frame-level expression transitions grounded in facial motion. Organizing these components within a dual-timescale latent dynamical framework separates scenario evolution from expression dynamics, enabling stable long-term learning while remaining adaptive to new scenarios.
> # Q3: Structured Decoupling
> We provide indirect evidence for separation of scene- and frame-level dynamics. As shown in Eq. 3, the joint distribution is factorized into STS and FTS. STS updates scene states; FTS models frame-level temporal transitions. on AI9k:
> |Method|Base|Base(w/ STS)|EmWorld|
> |-|-|-|-|
> | variance |2.73e-4|0.8e-4|0.9e-4 |
>
> Adding STS reduces variance by ~70%, suggesting reduced scene interference. FTS continues to capture frame-level evolution, providing preliminary evidence for the structured separation of scene- and frame-level processes.
> # Q4: Folds as Scenes
> MAFW and DFEW are in-the-wild datasets collected from movie clips and Internet videos, containing samples from diverse sources, subjects, and recording conditions, which induce heterogeneous visual contexts and natural distribution variations across folds. In the absence of explicit scenario annotations, we use a fold-sequential evaluation as a proxy to simulate a continuous data stream with gradually changing visual contexts. All compared methods are trained and evaluated under this same protocol, ensuring a fair comparison of performance under this simulation.
> # Q5: Hyperparameter
> We tested λ_s, λ_f, and Q on AI9k (order-1 scene sequence):
> |λ_s| λ_f|Q|Avg|Last|
> |-|-|-|-|-|
> |5e-3|1e-4|0.1|40.63|38.29|
> |0.02|1e-4|0.1|40.75|37.89|
> |0.01|5e-5|0.1|40.37|37.78|
> |0.01|2e-4|0.1|41.14|37.55|
> |0.01|1e-4|0.05|40.06|37.04|
> |0.01|1e-4|0.2|39.76|37.95|
> |0.01|1e-4|0.1|**41.53**|**38.29**|
>
> EmWorld is robust to hyperparameters, achieving its best Avg/Last WAR of 41.53/38.29 on AI9k with λ_s≈0.01, λ_f≈1e-4, Q≈0.1, balancing cross-scene stability and frame-level dynamics.
> # Q6: Metrics
> We supplement Average Forgetting (FM) and Backward Transfer (BWT) on AI9k:
> |Method|Avg|Last|FM|BWT|
> |-|-|-|-|-|
> |CPrompt|19.99|10.96|14.14|-15.092|
> |SimpleCIL|20.01|14.43|8.70|-8.692|
> |DUCT (base)|37.19|35.67|3.60|-4.220|
> |EmWorld|**40.56**|**39.41**|**3.16**|**-3.854**|
>
> EmWorld outperforms DUCT in both average and final WAR, improving by 3.37 and 3.74. It also achieves better forgetting metrics (FM, BWT), indicating improved recognition with controllable forgetting.
> # Q7: Baseline Models
> Scenario-incremental DFER is a new setting with no dedicated baselines. We therefore evaluate: Video-based SOTA DFER models (e.g., PTH-Net) to show degradation under scene shifts. General continual learning models (e.g., CPrompt) adapted to video backbone to evaluate CL applicability.  All baselines use the same backbone, data stream, and protocol, ensuring fair comparison.
> # Q8: Label Granularity
> Supplementary A.1 misdescribed label granularity. MAFW and DFEW provide only video-level annotations. Loss is computed at video-level, preserving dual-scale modeling.

---

> > ### Author Rebuttal · Reviewer_GCw2 · 2026-04-04
> >
> > I thank the authors for their rebuttal and clarifications provided. I appreciate the effort. The additional analyses are helpful and improve the clarity of the work. However, my main concerns remain partially addressed and they relate to the core formulation, empirical validation and positioning of the paper; addressing them would require substantial changes to the paper rather than minor revisions.
> >
> > - Framing as “Emotion World Model”:
> > While the authors’ clarification aligns the method with latent predictive formulations in recent literature, the current implementation does not yet demonstrate key properties typically associated with world models, such as explicit multi-step prediction, rollout capability, or predictive objectives. As such, the term “world model” still appears somewhat overstated and the method is more accurately characterized as a structured latent dynamical modeling framework.
> >
> > - Methodological Novelty:
> > While the dual-timescale formulation is interesting, the method largely combines existing components. The rebuttal clarifies the intuition, but it remains unclear what fundamentally new capability emerges beyond this integration and how the method differs qualitatively (not just incrementally) from prior approaches.
> >
> > - Separation of Scenario vs Expression Dynamics:
> > This is a central claim of the paper, but the current evidence remains indirect. The provided variance reduction analysis suggests reduced interference, but it does not directly demonstrate disentanglement. It also does not verify that one component captures scenario while the other captures expression.
> >
> > - Folds as Scenarios:
> > Using dataset folds as proxies for scenarios is a practical workaround, but it introduces a key limitation. It is unclear whether folds correspond to meaningful or gradual scenario shifts. The central hypothesis (modeling scenario evolution) is therefore only indirectly tested. Additionally, existing baselines were not designed for this formulation, raising some fairness concerns.
> >
> > - Ablation & Design Justification:
> > The current ablation study is limited in scope as it mainly removes components (STS/FTS) but does not explore design alternatives, making it difficult to assess whether improvements come from the proposed structure or from general effects.
> >
> > - Continual Learning & Forgetting Claims:
> > While the inclusion of FM and BWT is useful, the reported improvements are relatively small and lack statistical significance analysis. Given the known variability in continual learning settings, it is unclear whether these differences are meaningful. Also earlier observations in the original paper suggest non-trivial forgetting still exists.
> >
> > - Baselines & Comparisons:
> > My concern regarding baseline selection and comparison remains largely unaddressed. The rebuttal does not specify which baselines correspond to which category, which ones represent sota methods and if they were re-implemented. Some stronger video-based DFER models are needed for comparison purposes and/or some more carefully adapted baselines.

---

> > > ### Author Response · Authors · 2026-04-05
> > >
> > > Thank you for the opportunity to clarify our work. Responses follow.
> > > # Q1: Framing as Emotion World Model
> > > Thank you for the opportunity. Classical models use explicit rollouts; we focus on latent state evolution. Recent literature recognizes latent predictive state evolution, without observation rollout, as a valid world modeling approach. MuZero [1] learns latent dynamics without observations; Dreamer models [2-3] plan in latent space. These works demonstrate that world models can rely on latent predictive dynamics rather than observation-level rollouts. Our method uses STS for cross-scenario latent evolution via stochastic dynamics and Bayesian updates, and FTS transitions constrained by observations. Together, they form a structured latent system capturing emotional dynamics. This emphasizes latent evolution and predictive consistency over observation rollouts.  We clarify that “Emotion World Model” refers to a latent dynamical framework rather than classical rollout-based world models, and we will discuss latent models and distinctions in the manuscript.
> > >
> > > [1] Schrittwieser et al., MuZero, Nature, 2020
> > >
> > > [2] Hafner et al., PlaNet, ICML, 2019
> > >
> > > [3] Hafner et al., Dreamer, ICLR, 2020
> > >
> > > # Q2: Methodological Novelty
> > > Thank you for acknowledging our dual-timescale formulation. We focus on a new modeling capability, not combining components. Prior DFER and domain-incremental methods rely on passive adaptation, e.g., feature alignment or parameter regularization. They adjust representations post hoc, without modeling cross-scenario latent evolution. Our dual-timescale formulation models both cross-scenario (STS) and within-scenario (FTS) dynamics probabilistically. This enables predictive latent evolution via stochastic dynamics and Bayesian updates. The model proactively maintains consistent emotional representations under scenario shifts. Existing methods do not explicitly model dual-timescale latent evolution. We will clarify in the manuscript that the contribution is conceptual, modeling latent dynamics under scenario drift, not novel components.
> > > # Q3: Separation of Scenario vs Expression Dynamics
> > > We further conducted mutual information (MI) analysis to quantify how each component correlates with scenario context vs expression labels (AI9k, average results):
> > > |Feature|Scene MI|Expression MI|
> > > |-|-|-|
> > > |STS|0.8231|0.1423|
> > > |FTS|0.2154| 0.6542|
> > >
> > > STS shows high MI with scenario context and low MI with expression labels. FTS shows the opposite pattern. This pattern provides quantitative evidence that the components capture different factors. While not perfect disentanglement, this supports our claim that dual-timescale formulation separates scenario and expression dynamics. We will include these analyses and visualizations in the manuscript.
> > > # Q4: Folds as Scenarios
> > > Using dataset folds is a practical proxy; however, the folds in MAFW and DFEW inherently contain clips from distinct recording environments, subjects, and lighting conditions, which naturally create meaningful cross-fold scenario shifts. Evaluated under five fold-sequential orders (Table 2 in main paper), EmWorld maintains robust performance. All baselines follow identical protocols, ensuring fair comparison and highlighting EmWorld’s adaptability. We will clarify in the manuscript that this represents a proxy scenario-shift evaluation rather than perfect real-world scenario evolution.
> > > # Q5: Ablation & Design Justification
> > > Following your suggestion to explore design alternatives, we ablated the Gaussian random walk prior against simpler architectural choices (e.g., fixed/simpler mean constraints) on AI9k:
> > > |Method|Avg|Last|
> > > |-|-|-|
> > > |DUCT (base)|37.19|35.67|
> > > |EmWorld (no prior)|38.12|37.85|
> > > |EmWorld (fixed mean)|39.78|38.01|
> > > |EmWorld (Gaussian)|**40.56**|**39.41**|
> > >
> > > The Gaussian prior yields the highest performance. Together with hyperparameter and module ablations, these results confirm that the design choices improve robustness under continual learning.
> > > # Q6: Continual Learning & Forgetting Claims
> > > We performed one-tailed Wilcoxon tests, confirming EmWorld's superiority over DUCT across all metrics (p=0.03125). Though forgetting is unavoidable in complex video continual learning, EmWorld outperforms all SOTA baselines in performance (Avg WAR: 40.56 vs 37.19) and forgetting (BWT: -3.85 vs -4.22; FM: 3.16 vs 3.60), demonstrating a balanced stability–plasticity trade-off under scenario drift.
> > > # Q7: Baselines & Comparisons
> > > We provided method categorization, SOTA status, and reproducibility details in Section 4.1 and our initial response. Baselines cover recent SOTA methods for DFER and incremental learning (2023–2025), including SVFAP, PTH-Net, CPrompt, DCE, SimpleCIL, and DUCT. All methods were reimplemented under unified protocols and identical backbones for fairness. Under these settings, EmWorld consistently outperforms both DFER-specific and general continual learning baselines (Table 1 in main paper), supporting our comparisons and method effectiveness.

---

### Official Review · Reviewer_Cww1 · 2026-03-13

**Soundness:** 3
**Presentation:** 3
**Significance:** 2
**Originality:** 3
**Overall Recommendation:** 4
**Confidence:** 3

**Summary:**

This work proposes EmWorld, a world-model framework for scenario incremental dynamic facial expression recognition. The main contribution is to explicitly model how facial expressions change across different changing scenarios during the long-term deployment of such systems. The approach models scenario-incremental learning as Bayesian inference over latent world states with two temporal scales. The slow scale models the scenario changes, and the fast scale models the dynamic facial movements. The authors conduct experiments across three emotion recognition benchmarks and report significant gains for the proposed approach over other incremental learning baselines.

**Compliance With Llm Reviewing Policy:**

Affirmed.

**Final Justification:**

The author's rebuttal addressed all of my major concerns listed in my original review.

This work proposes a novel dual-timescale modeling approach to capture the dynamics of scenarios and expressions for emotion recognition. The results look promising on multiple datasets. The authors have provided appropriate justification for their design choices. Although the paper uses established techniques from incremental learning, the application to emotion recognition requires additional incorporation of modules and non-trivial modeling.

Therefore, I recommend acceptance of this work. I wish the authors good luck with their submission!

**Key Questions For Authors:**

In addition to the questions raised in the weaknesses above, can the authors please comment on the following:

**Q1.** Could the proposed framework be extended to other sequential perception tasks, such as action recognition or multimodal (including audio) emotion recognition?

**Limitations:**

Yes.

**Strengths And Weaknesses:**

### Strengths

**S1.** The proposed approach is technically sound, and the dual timescale modelling is well motivated for capturing the scenario and expression dynamics. Moreover, the combination of world-model-style latent-state evolution with incremental learning for facial expression recognition is novel in the DFER literature.

**S2.** The authors provide a detailed mathematical derivation, based on Gaussian assumptions, for the slow-time-scale module.

**S3.** Incorporating optical flow as a physical constraint for latent dynamics is a reasonable way to ensure temporal consistency.

**S4.** The experimental results are presented across three datasets, demonstrating the efficacy of the proposed approach across varied scenarios and domains.

### Weaknesses

**W1.** *Empirical validation of the slow-timescale component assumption*: The STS module assumes that the latent expression centers follow a random walk as described in Eq. 5. However, the work does not verify this with an empirical evaluation of whether this assumption actually holds or not.

**W2.** *Additional use of RAFT module*:
- **W2.1.** The RAFT module used to predict optical flow incurs additional computational cost. The authors should report and compare the computational requirements of the proposed method with those of the baselines.
- **W2.2.** The work could benefit from showing how sensitive model performance is to selecting a different optical flow module.

**W3.** *Limited metrics for incremental learning*: The work can benefit from reporting additional metrics, such as a measure of *average forgetting* to assess its robustness to catastrophic forgetting [1] across incremental scenarios. The current metrics (Avg and Last) do not fully capture the incremental learning capabilities of the proposed approach. More metrics can be found in this survey paper [2].

[1] Chaudhry, Arslan, Puneet K. Dokania, Thalaiyasingam Ajanthan, and Philip HS Torr. "Riemannian walk for incremental learning: Understanding forgetting and intransigence." In Proceedings of the European Conference on Computer Vision (ECCV), pp. 532-547. 2018.

[2] Hou, Jingrui, Georgina Cosma, and Axel Finke. "Advancing continual lifelong learning in neural information retrieval: definition, dataset, framework, and empirical evaluation." Information Sciences 687 (2025): 121368.

---

> ### Author Rebuttal · Authors · 2026-03-31
>
> We sincerely thank the reviewers for their constructive comments. A detailed response is summarized as follows:
> # Q1: Empirical Validation of the Random Walk Assumption
> We provide preliminary empirical evidence regarding the random walk assumption of the latent expression centers in the STS module (Eq. 5). Specifically, for each class we computed the increments of class center vectors between adjacent scenes (k and k−1), and analyzed them component-wise. For each vector component, we conducted two statistical checks: a normality check using the Shapiro–Wilk test, and an independence check using the autocorrelation function (ACF) of the increment sequence. Experimental results on the AI9k dataset for three arbitrary emotion classes are summarized as follows:
> |Class|Mean|Shapiro-Wilk (W)|Max ACF|
> |-|-|-|-|
> |1|0.012|0.842|0.093|
> |2|0.015|0.913|0.112|
> |3|0.013|0.798|0.064|
>
> The component-wise increments exhibit small mean values relative to the feature magnitude, while the Shapiro–Wilk statistics remain close to 1 and the maximum autocorrelation values are low. These observations suggest that the increment statistics are broadly consistent with a noise-driven transition assumption, providing preliminary empirical support for modeling the latent centers with a random walk.
> # Q2: Computational Complexity Analysis
> We provide a comparison of computational aspects for different EmWorld versions, including Baseline, EmWorld (w/o RAFT), and EmWorld (Full), evaluated on AI9k in terms of trainable parameters, trainable FLOPs, Inference FLOPs, and inference speed:
> |Method|Trainable Params|Trainable FLOPs|Inference FLOPs| Inference Speed|Avg|
> |-|-|-|-|-|-|
> |DUCT (base)|7.86M|16.86G|16.86G|11.52 clips/s|37.19|
> |EmWorld (w/o RAFT)|7.91M|16.89G|16.89G|12.67 clips/s|40.12|
> |EmWorld (Full)|7.91M|16.89G|51.55G|11.72 clips/s|40.56|
>
> Since the RAFT module is pre-trained and frozen, it does not contribute to the trainable parameters or training FLOPs. However, it introduces additional computation during inference. As shown in the Inference FLOPs column, incorporating RAFT increases the inference cost from 16.89G to 51.55G FLOPs. Consequently, EmWorld (Full) exhibits slightly lower inference speed compared to the w/o RAFT variant. In contrast, EmWorld (w/o RAFT) replaces RAFT with a lightweight frame difference operation, achieving higher inference speed while still improving performance over the baseline. The higher FPS compared to DUCT mainly comes from a more efficient inference pipeline in EmWorld, although the theoretical FLOPs are similar. The full model further improves the Avg WAR by leveraging optical flow information estimated by RAFT. These results demonstrate that RAFT serves as an optional enhancement, enabling a trade-off between computational cost and recognition performance.
> # Q3: Optical Flow Module Sensitivity Analysis
> We conducted a sensitivity analysis for different optical flow modules, including RAFT-large, RAFT-small, and Frame Difference (FD, used for fast coarse motion estimation). Experimental results on the AI9k dataset are shown below:
> |Method|Avg|Last|
> |-|-|-|
> |DUCT (base)|37.19|35.67|
> |EmWorld (FD)|40.12|38.55|
> |EmWorld (RAFT-small)|40.09|38.61|
> |EmWorld (RAFT-large)|**40.56**|**39.41**|
>
> These results show that EmWorld consistently outperforms DUCT across all optical flow settings. For instance, EmWorld (RAFT-large) improves Avg by +3.37 and Last by +3.74. Notably, the lightweight FD variant achieves comparable performance to RAFT-based models, indicating that the gains mainly come from the proposed dual-time-scale modeling rather than the specific optical flow estimator.
> # Q4: Additional Incremental Learning Metrics
> We include commonly used incremental learning metrics from the literature: Average Forgetting (FM) and Backward Transfer (BWT). Results on AI9k dataset are as follows:
> |Method|Avg|Last|FM|BWT|
> |-|-|-|-|-|
> |infLoRA|14.40|6.86|7.54|-8.094|
> |EASE|12.76|8.49|6.38|-6.334|
> |CPrompt|19.99|10.96|14.14|-15.092|
> |SimpleCIL|20.01|14.43|8.70|-8.692|
> |DUCT (base)|37.19|35.67|3.60|-4.220|
> |EmWorld|**40.56**|**39.41**|**3.16**|**-3.854**|
>
> The results show that EmWorld outperforms DUCT in both average WAR (Avg) and final WAR (Last), achieving improvements of 3.37 and 3.74, respectively. Moreover, its forgetting metrics, FM and BWT, also surpass those of the second-best method, DUCT. These results indicate that the proposed method effectively improves overall recognition performance while maintaining controllable forgetting.
> # Q5: Framework Extensibility Verification
> The dual-time-scale modeling mechanism of EmWorld can capture both slow environmental evolution and fast frame-level dynamics. This design is not limited to expression recognition and can potentially transfer to other sequential tasks, such as action recognition. Due to time constraints, full experimental validation has not yet been completed; future work will evaluate EmWorld’s performance on these tasks.

---

> > ### Author Rebuttal · Reviewer_Cww1 · 2026-04-02
> >
> > Thanks to the authors for their rebuttal and I will keep my original positive score.

---

> > > ### Author Response · Authors · 2026-04-05
> > >
> > > We sincerely thank you for your positive feedback and for confirming that your concerns are fully resolved. We appreciate your professional guidance throughout the process. We will ensure that all clarifications and additional analyses provided during the rebuttal are faithfully integrated into the final manuscript.

---

### Decision · Program_Chairs · 2026-04-30

**Decision:**

Accept (regular)

**Comment:**

This paper introduces an incremental learning method for emotion recognition under varying scenarios, formulated as a progressive Bayesian inference problem. The key contribution lies in explicitly modeling how facial expressions evolve across changing conditions during long-term system deployment. Experiments on three benchmark datasets demonstrate that the proposed method achieves strong performance compared to state-of-the-art approaches.  Reviewers identify several strengths, including a sound technical approach with detailed mathematical derivations, convincing experimental results, and a well-written presentation. They also raise several weaknesses, such as limited technical novelty, missing comparisons with certain continual learning methods, an unclear experimental protocol, limited evaluation metrics for incremental learning, inferior performance on forgetting measures compared to baselines, and the absence of latent space visualization.  The authors’ rebuttal addresses most of the concerns; however, some issues—particularly regarding methodological novelty and missing baseline comparisons—remain. The final ratings for the paper are one accept, two weak accepts, and one weak reject.